# Non-Linear Operator Approximations for Initial Value Problems

**Gaurav Gupta** [1]**, Xiongye Xiao** [1]**, Radu Balan** [2]**, Paul Bogdan** [1]

[1] Ming Hsieh Department of Electrical and Computer Engineering
University of Southern California
Los Angeles, CA 90089, USA
[2] Department of Mathematics and the Norbert Wiener Center for Harmonic Analysis and Applications
University of Maryland
College Park, MD 20742, USA
{ggaurav, xiongyex, pbogdan}@usc.edu, rvbalan@math.umd.edu,

## Abstract

Time-evolution of partial differential equations is fundamental for modeling several complex dynamical processes and events forecasting, but the operators associated with such problems are non-linear. We propose a *Padé approximation based exponential neural operator* scheme for efficiently learning the map between a given initial condition and the activities at a later time. The multiwavelets bases are used for space discretization. By explicitly embedding the exponential operators in the model, we reduce the training parameters and make it more data-efficient which is essential in dealing with scarce and noisy real-world datasets. The Padé exponential operator uses a *recurrent structure with shared parameters* to model the non-linearity compared to recent neural operators that rely on using multiple linear operator layers in succession. We show theoretically that the gradients associated with the recurrent Padé network are bounded across the recurrent horizon. We perform experiments on non-linear systems such as Korteweg-de Vries (KdV) and Kuramoto–Sivashinsky (KS) equations to show that the proposed approach achieves the best performance and at the same time is data-efficient. We also show that urgent real-world problems like epidemic forecasting (for example, COVID-19) can be formulated as a 2D time-varying operator problem. The proposed Padé exponential operators yield better prediction results (**53%** (**52%**) better MAE than best neural operator (non-neural operator deep learning model)) compared to state-of-the-art forecasting models.

## 1 Introduction

Predicting the future states using the current conditions is a fundamental problem in machine learning, robotics, autonomous aerial / ground / underwater systems and cyber-physical systems (Xue & Bogdan (2017)). Such problems fall under the umbrella of a common term, the "Initial Value Problems" (IVPs). The basic structure of IVP involves a first-order time-evolution along with non-linear operators. The class of IVPs spans the domain of physics (modeling gravitational waves (Lovelace, 2021)), neuroscience (Hodgkin-Huxley model (Zhang et al., 2020)), engineering (fluid dynamics (Wendt, 2008)), water waves (tsunami (Elbanna et al., 2021)), mean field games (Ruthotto et al., 2020; Bogdan & Marculescu, 2011), to list just a few. Within the current pandemic context, the applications areas like epidemiology (Kermack–McKendrick model (Kermack et al., 1991; Diekmann et al., 2021)) are of tremendous interest.

**Neural Operators** The use of deep learning to solve the IVP like problems for predictions has been exploiting within the framework of convolutional neural networks (CNNs) (Bhatnagar et al., 2019; Guo et al., 2016), and time-evolution by employing multiple layers (Khoo et al., 2020). The multi-layered deep networks with CNNs are suitable to solve problems with a large number of training samples. Moreover, because of the image-regression like structures, such models are restricted to the specifications of the input size. Another research direction aims at solving and modeling the partial

differential equations (PDEs) versions of the IVPs for a given instance. The works of (Kochkov et al., 2021) model the IVP solution as NNs for modeling the turbulent flows. Along the same lines, we have physics-informed neural networks (PINNs) (Raissi et al., 2019; Wang et al., 2021b) that utilize PDE structure for defining the loss functions. Such models are not applicable within the context of a complete data-driven scenario, or for the setups where the exact PDE structure is not known, for example, modeling the climate, epidemic, or unknown physical and chemical phenomena. Finally, we have the works of *Neural Operators* that are completely data-driven and input-resolution independent schemes (Li et al., 2020b;c;a; Gupta et al., 2021; Bhattacharya et al., 2020; Patel et al., 2021). Most of these approaches tried to efficiently work with the integral kernel operators, for example, Graph Nyström sampling in (Li et al., 2020b), convolution approximation in (Li et al., 2020a), or the multiwavelets compression in (Gupta et al., 2021). Apart from solving non-homogeneous linear differential equations, the PDE operators are mostly non-linear. To tackle the non-linear behavior, these prior works use a multi-cell architecture with non-linearity (for example, ReLU). To work with the IVP like problems, and be data-efficient, we aim to adopt explicitly the non-linear operator (exponential) that appears in the IVP solutions.

**Exponential Operators**  The exponential of linear transformation has been a subject of research for the last 150 years (Laguerre, 1898). In the simplest form, the exponential operator appears as a solution of: $\frac{dy}{dt} = at, y(0) = y_0$ as $y(t) = e^{at}y_0$ (for more general examples, see Table 1). With applications ranging from control systems theory (converting continuous to discrete systems) to solving partial differential equations (Cox & Matthews, 2002; Kassam & Trefethen, 2005), the exponential function of operators is a subject of active research. In deep learning, the exponential function to model non-linearity is used in (Andoni et al., 2014). Recently, the exponential operators have also been explored in the field of computer vision for generative flows in (Hoogeboom et al., 2020).

**Padé Approximation**  Although one approach to implementing an exponential operator could be attained through a Taylor series representation, this operator function is prone to errors (Abramowitz & Stegun, 1965). Scale-and-squaring (SSQ) methods are commonly suggested approaches to deal with the errors (Lawson, 1967). In addition to SSQ, the Padé approximation which represents an analytic function as the ratio of polynomials achieves state-of-the-art accuracy in computing exponential operators (Fasi & Higham, 2019). Industry standard numerical toolboxes (for example, MATLAB, SciPy) use the Padé approximation based approach to compute the matrix exponential `expm` (Al-Mohy & Higham, 2009). Matrix exponential via Padé representation requires dense matrix computations (for example, inverse and higher-order polynomials). Such operations are not numerically feasible, in-general, for the inputs with large size. However, the commonly used operators like convolution (possibly, multi-layered) have parameters that are fixed beforehand and are much less than the input dimension. A suitable approach, therefore, is a neural architecture based Padé approximation.

Our strategy, in this work, is to explicitly embed the exponential operators in the neural operator architecture for dealing with the IVP like datasets. The exponential operators are non-linear, and therefore, this removes the requirement of having multi-cell linear integral operator layers. While with sufficient data in-hand, the proposed approach may work similarly to the existing neural operators with a large number of training parameters. However, this is seldom a feasible scenario for the expensive real-world experiments, or on-going recent issues like COVID19 prediction. Here, the current work is helpful in providing data-efficiency analytics, and is useful in dealing with scarce and noisy datasets (see Section 3.3). To the advantage of Padé approximation, the exponential of a given operator can be computed with the pre-defined coefficients (see Section 2.3) and a recurrent polynomial mechanism.

**Our Contributions**  The main novel contributions of this work are summarized as follows: (*i*) For the IVPs, we propose to embed the exponential operators in the neural operator learning mechanism. (*ii*) By using the Padé approximation, we compute the exponential of the operator using a novel recurrent neural architecture that also eliminates the need for matrix inversion. (*iii*) We theoretically demonstrate that the proposed recurrent scheme, using the Padé coefficients, have bounded gradients with respect to (w.r.t.) the model parameters across the recurrent horizon. (*iv*) We demonstrate the data-efficiency on the synthetic 1D datasets of Korteweg-de Vries (KdV) and Kuramoto–Sivashinsky (KS) equations, where with less parameters we achieve state-of-the-art performance. (*v*) We formulate and investigate the epidemic forecasting as a 2D time-varying neural operator problem, and show

| Equation | Solution |
|---|---|
| $\frac{d\mathbf{u}}{dt} = \mathbf{A}\mathbf{u}$, $u(t=0) = \mathbf{u}_0$ (Linear ODE) | $\mathbf{u}(t=\tau) = e^{t\mathbf{A}}\mathbf{u}_0$ |
| $u_t = \frac{\partial^2 u}{\partial x^2}$, $u(x,0) = u_0(x)$ (Heat equation) | $u(x,\tau) = e^{\tau \frac{\partial^2}{\partial x^2}} u_0(x)$ |
| $u_t = \mathcal{L}u + \mathcal{N}f(u)$, $u(x,0) = u_0(x)$ [1] | $u(x,\tau) = e^{\tau\mathcal{L}}u_0(x) + \int_0^\tau e^{(\tau-t)\mathcal{L}}\mathcal{N}f(u(x,t))dt$ [2] |

Table 1: Initial Value Problem examples along with their time-evolution solutions. The exponential function of operators appears in the IVP solutions. For Linear ODE, $\mathbf{A}$ is the linear transformation.

that for real-world noisy and scarce data, the proposed model outperforms the best neural operator architectures by 53% and best non-neural operator schemes by 52%.

## 2 OPERATORS FOR INITIAL VALUE PROBLEM

We formalize the partial differential equations (PDEs) version of the Initial Value Problem studied in this work in Section 2.1. Section 2.2 summarizes the multi-resolution analysis using multiwavelets for space-discretization. Section 2.3 describes the proposed use of canonical exponential operators and presents a novel architecture using Padé approximation.

### 2.1 INITIAL VALUE PROBLEM

The initial value problem (IVP) for PDEs can be written in its general form as follows.

$$u_t = \mathcal{F}(t, u), \quad x \in \Omega$$
$$u(x, 0) = u_0(x), \quad x \in \Omega \tag{1}$$

where, $u_t$ is the first-order time derivative of $u$, $\mathcal{F}$ is a time-varying differential operator (non-linear in-general) such that $\mathcal{F} : \mathbb{R}^+ \cup \{0\} \times \mathcal{B} \to \mathcal{B}$ with $\mathcal{B}$ being a Banach space. Usually, the system in eq. (1) is required to satisfy a boundary condition such that $Bu(x, t) = 0$, $x \in \partial\Omega$ $\forall t$ in the solution horizon, and $\partial\Omega$ is the boundary of the computational region $\Omega$ with $B$ some linear function. Pertaining to our work, the operator map problem for IVP can be formally defined as follows.

**Operator Problem** Given $\mathcal{A}$ and $\mathcal{U}$ as two Sobolev spaces $\mathcal{H}^{s,p}$ with $s > 0, p = 2$, an operator $T$ is such that $T : \mathcal{A} \to \mathcal{U}$. For a given $\tau > 0$ and two functions $u_0(x)$ and $u(x, \tau)$, in this work, we take the operator map as $T\,u_0(x) = u(x, \tau)$ with $x \in \Omega$.

Table 1 summarizes a few examples of the IVP and their solutions. The exponential operators are ubiquitous in the IVP solutions and, therefore, are important to study. One issue, however, is that the exponential operators are non-linear and unlike convolution like operators, there does not exist a *general way* to diagonalize them (Fourier transform diagonalizes convolution operator) for an efficient representation. Previous work on neural operators (Li et al., 2020c;a; Gupta et al., 2021) modeled the non-linear operators in one way or another by using multiple canonical integral operators along with non-linearity (for example, ReLU). In this work, we directly produce an exponential operator approximation. First, we discuss an efficient basis (multiwavelets) for space discretization of the input / output functions in Section 2.2.

### 2.2 MULTI-RESOLUTION ANALYSIS

The multi-resolution analysis (MRA) aims at projecting a function to a basis over multiple scales. The wavelet basis (e.g., Haar, Daubechies) are some popular examples. Multiwavelets further this operation by using the family of orthogonal polynomials (OPs), for example, Legendre polynomials for an efficient representation over a finite interval (Alpert et al., 2002). The multiwavelets are

---

[1]Time-advection equation with linear operators $\mathcal{L}, \mathcal{N}$ and non-linear function $f(.)$. A wide range of problems can be modeled, for example, Korteweg-de Vries, Kuramoto-Sivashinsky, Burgers' Equation, Navier-Stokes (list not exhaustive).

[2]A non-linear integro-differential solution to the time-advection equation using semi-group approach (Beylkin & Keiser, 1997; Pazy, 1983; Yoshida, 1980). A slightly general version is discussed in (Beylkin et al., 1998).

useful in the sparse representation of the integral operators with smooth kernels. In addition, the multiwavelets also sparsify the exponential functions of the strictly elliptic operators (Beylkin & Keiser, 1997). However, we do not rely on this assumption in this work. Here, we briefly introduce the MRA and refer the reader to Gupta et al. (2021) for a more detailed discussion.

**Notation** We begin by defining the space of finite interval polynomials as $\mathbf{V}_n^k = \{f | f$ are polynomials of degree $< k$ defined over interval $(2^{-n}l, 2^{-n}(l+1))$ for all $l = 0, 1, \ldots, 2^n - 1$, and assumes 0 elsewhere$\}$. The $\mathbf{V}_n^k$ are contained in each other for subsequent $n$ or,

$$\mathbf{V}_0^k \subset \mathbf{V}_1^k \subset \ldots \subset \mathbf{V}_{n-1}^k \subset \mathbf{V}_n^k \subset \ldots. \tag{2}$$

The orthogonal component of these polynomial spaces is termed as multiwavelet space $\mathbf{W}_n^k$ and are defined such that

$$\mathbf{V}_n^k \bigoplus \mathbf{W}_n^k = \mathbf{V}_{n+1}^k, \quad \mathbf{V}_n^k \perp \mathbf{W}_n^k. \tag{3}$$

The orthonormal basis of $\mathbf{V}_0^k$ are OPs $\phi_0, \phi_1, \ldots, \phi_{k-1}$ and we have used appropriately normalized shifted Legendre Polynomials in this work. The basis for $\mathbf{V}_n^k$ and $\mathbf{W}_n^k$ are $\phi_{jl}^n(x) = 2^{n/2}\phi_j(2^n x - l)$ and $\psi_{jl}^n(x) = 2^{n/2}\psi_j(2^n x - l)$, respectively, for $l = 0, 1, \ldots, 2^n - 1$ and $j = 0, 1, \ldots, k - 1$.

Finally, an important trick for representing the operator $T$ in the multiwavelet basis is called non-standard (NS) form (Beylkin et al., 1991). The NS form decouples the interactions of the scales and is useful in obtaining an efficient numerical procedure. Using NS form, the projection of operator $T$ is expanded using a telescopic sum as follows.

$$T_n = \sum_{i=L+1}^n (Q_i T Q_i + Q_i T P_{i-1} + P_{i-1} T Q_i) + P_L T P_L, \tag{4}$$

where, $P_n : \mathcal{H}^{s,2} \to \mathbf{V}_n^k$ is the projection operator, $T_n = P_n T P_n$, $Q_n : \mathcal{H}^{s,2} \to \mathbf{W}_n^k$ such that $Q_n = P_n - P_{n-1}$, and $L$ is the coarsest scale under consideration ($L \geqslant 0$). Therefore, the NS form of the operator is a collection of the triplets $\{A_i, B_i, C_i\}_{i=L+1}^n$ and $P_L T P_L$ with $A_i = Q_i T Q_i, B_i = Q_i T P_{i-1}$ and $C_i = P_{i-1} T Q_i$. In this work, we aim to model $A_i, B_i, C_i$ as the exponential operators to better learn the IVP by explicitly embedding the non-linear operators into the multiwavelet transformation. This is not straightforward due to the non-linearity of exponential functions. We are now in shape to present the main contribution of the current work in the Section 2.3 where we discuss an implementable neural approximation of the exponential operators.

## 2.3 EXPONENTIAL OPERATOR APPROXIMATIONS

Due to the nature of first-order time-evolution equations, the exponential operators appear in the solution of IVP as discussed in Section 2.1. Being an analytic function, the exponential also assumes a Taylor series expansion. However, the approximation error by truncation is (Abramowitz & Stegun, 1965) and may require a large number of coefficients. We now discuss a better approximation for the non-linear functions.

**Padé Approximation** Given an analytic function $f(z), z \in \mathbb{C}$ at 0 and let $p, q \in \mathbb{N}$, the $[p/q]$ Padé approximation of $f$ at 0 is a rational polynomial $r_{pq}(z) = A_{pq}(z)/B_{pq}(z)$, where $A_{pq}(z)$ and $B_{pq}(z)$ are polynomials of degree $p$ and $q$, respectively. For our work, $f(z) = e^z$, and the $[p/q]$ Padé approximation to the exponential function is written as $A_{pq}(z) = \sum_{j=0}^p a_j z^j$ and $B_{pq}(z) = \sum_{j=0}^q b_j z^j$, where

$$a_j = \frac{(p+q-j)!p!}{(p+q)!j!(p-j)!}, 0 \leqslant j \leqslant p \quad b_j = \frac{(p+q-j)!q!(-1)^j}{(p+q)!j!(q-j)!}, 0 \leqslant j \leqslant q. \tag{5}$$

Note that, $a_0 = b_0 = 1$, and we now discuss the exponential of operators. Given an operator $\mathcal{L}$ (possibly non-linear), the $[p/q]$ Padé approximation for $e^{\mathcal{L}}$ can be written as

$$e^{\mathcal{L}} \approx r_{pq}(\mathcal{L}) = \left( \sum_{j=0}^q b_j \underbrace{\mathcal{L} \circ \mathcal{L} \circ \ldots \circ \mathcal{L}}_{j-\text{times}} \right)^{-1} \left( \sum_{j=0}^p a_j \underbrace{\mathcal{L} \circ \mathcal{L} \circ \ldots \circ \mathcal{L}}_{j-\text{times}} \right). \tag{6}$$

For $\mathcal{L}$ being a linear transformation, the inverse in eq. (6) is computed as a matrix inverse, for example when evaluating a matrix exponential. Even convolution operator can be represented as the circulant

matrix (shift-invariant kernel) operation. But storing the big circulant matrix is not a numerically efficient solution. Instead, we now discuss a neural architecture that emulates the Padé approximation for operators in eq. (6) while avoiding taking the inverse.

**Padé Exponential Model** We compute the operator exponential using the $[p/q]$ Padé approximation via a recurrent neural architecture in Figure 1. First, the denominator polynomial $B_{pq}(\mathcal{L})$ is evaluated using the left recurrent network. Second, the output is passed through a non-linear layer which we implement as $v = \sigma(Wu + b)$ with $\sigma(.)$ being the ReLU function. Note that the non-linearity layer is applied to the input channels, and therefore, its size is independent of the input spatial dimension. Finally, the output of the non-linear layer is passed through another recurrent network implementing the numerator polynomial $A_{pq}(\mathcal{L})$. Note that, both polynomial recurrent networks use same operator $\mathcal{L}$, or in other words, the parameters are shared across the network. With $a_j, b_j$ fixed-beforehand, the total trainable parameters of the Padé exponential model are $\{\theta_{\mathcal{L}}, W, b\}$.

One issue with the recurrent architectures is the possibility of gradients explosion when evaluated over a large horizon. In such cases, techniques like gradient clipping is used as a workaround. For a given $p, q$, the proposed network runs the recurrent loops for a total of $p + q - 2$. We show that the proposed $[p/q]$ Padé network does not suffer from the issue of gradient explosion, and the boundedness of the gradients is established through the following result:

**Theorem 1.** *Given a linear operator $\mathcal{L} = \mathcal{L}(\theta_{\mathcal{L}})$, a non-linearity layer $v = \sigma(Wu + b)$, and $p, q \in \mathbb{N}$, at points of differentiability, the gradients of the operation $x \mapsto y = F(x; \theta_{\mathcal{L}}, W, b) := [p/q]e^{\mathcal{L}}(x)$ using the $[p/q]$ Padé network in Figure 1 are bounded in operator norm by*

$$\left\| \frac{\partial y}{\partial \theta_{\mathcal{L}}} \right\| \leqslant \exp(\|\mathcal{L}\|) \left( \|b\|_2 + \|W\| \|x\|_2 \right) \left( \sum_{j=1}^{n_\theta} \left\| \frac{\partial \mathcal{L}}{\partial \theta_j} \right\|^2 \right)^{1/2}, \quad (7)$$

$$\left\| \frac{\partial y}{\partial W} \right\| \leqslant \exp(\|\mathcal{L}\|) \|x\|_2, \quad (8)$$

$$\left\| \frac{\partial y}{\partial b} \right\| \leqslant \exp\left( \frac{p}{p+q} \|\mathcal{L}\| \right). \quad (9)$$

The detailed proof is provided in the Supplementary Materials (Appendix E). Note that, $n_\theta = |\theta_{\mathcal{L}}|$ is such that $n_\theta \ll M$, where $M$ is the dimension of the input. For example, the convolution operator has a fixed kernel independent of the size of the input, the Fourier-based convolution in (Li et al., 2020a) has $k_m$ Fourier modes independent of the input dimension. Next, as noted earlier, the non-linearity layer $\sigma(Wu + b)$ is applied to the input channels (instead of input spatial dimensions), therefore, the gradient bounds do not scale with high input-resolutions, apart from the dependence from $\|x\|_2$ which could be pre-normalized.

Finally, the Padé neural model is integrated with the multiwavelet transform by substituting each of $A, B, C$, from Section 2.2, with $[p/q]e^{\mathcal{L}_A}, [p/q]e^{\mathcal{L}_A}, [p/q]e^{\mathcal{L}_A}$. The complete flow-diagram by plugging-in the Padé model is shown in Appendix D.

## 3 EXPERIMENTS

Here, we empirically evaluate the proposed model and compare against the existing approaches. We consider several synthetic PDE datasets as well as a real-world example of pandemic prediction (COVID-19). For the data, a similar input/output structure is used as in the recent works of neural operator architectures. Specifically, the input function $u_0(x)$ and the output function $u(\tau, x)$ for some pre-specified $\tau > 0$, are evaluated at $M$ discretized locations of the domain $\Omega$. This yields a single training sample of $(u_0(x_i), u(\tau, x_i), x_i \in \Omega, 1 \leqslant i \leqslant M$. In total, we take $N$ training samples, and unless stated otherwise, $N = 1000$ and we test on 200 samples for the synthetic datasets.

**Padé Model Implementation** The operator $\mathcal{L}$ in Figure 1 is fixed as a single-layered convolution operator for 1D datasets, and 2-layered convolution for 2D datasets. For getting the input/output operator mapping, the multiwavelet transform is used only for discretizing the spatial domain. The Padé neural model easily fits into the sockets of the multiwavelet transformation based neural operator as sketched in Figure 7 (Appendix D). The multiwavelet filters are obtained using shifted Legendre OPs with degree $k = 4$. In contrast to the work of Gupta et al. (2021), only a single cell of

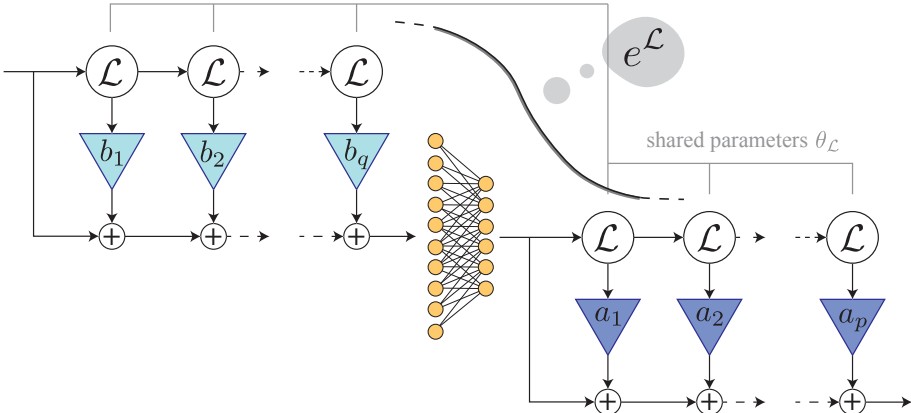

Figure 1: **Padé Exponential Model**. A recurrent neural architecture for computing the exponential of an operator $\mathcal{L}$ using $[p/q]$ Padé approximation. The multiplicative scalar coefficients $a_i, b_i$ are fixed-beforehand using eq. (5). The non-linear fully-connected layer is used to mimic the inverse polynomial operation.

multiwavelet transform is used in the current work because the non-linearity in the operator is explicit via Padé neural operator. This saves a lot of trainable parameters and yields a compact model which is suitable when dealing with scarce noisy real-world data as we see in Section 3.3. The numerator / denominator polynomials degrees $p/q$ for the Padé approximation is fixed as $(p, q) = (5, 6)$ for 1D models and $(4, 2)$ for 2D models. While the authors do not advocate that this is the best possible choice, an ablation study for 1D data is performed in Appendix B.1.

**Benchmark Neural Operators**  We compare against the recently proposed neural operator works with the proposed Padé exponential model (**Padé Exp**). The graph neural operator (**GNO**) was proposed in (Li et al., 2020b). A multi-level version of the graph neural network (**MGNO**) in (Li et al., 2020c). **LNO** A low-rank representation of the integral operator kernel and then using multiple layers with non-linearity, which also emulates unstacked DeepONet (Lu et al., 2020). A convolution approximation to the canonical integral kernel and then Fourier transform to diagonalize in (Li et al., 2020a) as **FNO**. **MWT Leg** utilize multiwavelets for spatial projections using Legendre OPs and uses multi-cell structure along with ReLU non-linearity. The **Padé Exp** model delivers state-of-the-art (Sota) performance on a range of datasets, both synthetic and real-world. With less number of parameters, the proposed model shows promise for small datasets as shown in Section 3.3.

**Training parameters**  All neural operator models are trained using Adam optimizer with a learning rate of $0.001$ and decay of $0.95$ after every $100$ steps. The loss function is taken as the relative L2 error. For synthetic datasets we train for a total of $500$ epochs and for real-world COVID-19 dataset we train for a total of $750$ epochs. All experiments are done on an Nvidia $A100$ 40GB GPUs.

## 3.1 KORTEWEG-DE VRIES EQUATION

The Korteweg-de Vries (KdV) equation is a one-dimensional non-linear PDE used to model the non-linear shallow water waves. For a given field $u(x, t)$, the KdV PDE takes the following form:

$$u_t = -0.5u\frac{\partial u}{\partial x} - \frac{\partial^3 u}{\partial x^3}, x \in (0, 1), t \in (0, 1]$$
$$u_0(x) = u(x, t = 0). \tag{10}$$

For Table 1 notations, we have $\mathcal{L} = -\frac{\partial^3}{\partial x^3}$, $f(u) = -0.25u^2$, and $\mathcal{N} = \frac{\partial}{\partial x}$. The neural operator learns the mapping of the initial condition $u_0(x)$ to the solutions $u(x, t = 1)$. The initial condition is generated in Gaussian random fields according to $u_0 \sim \mathcal{N}(0, 7^4(-\Delta + 7^2 I)^{-2.5})$ with periodic boundary conditions. The data set is obtained by solving the equation using the fourth-order stiff time-stepping scheme known as *ETDRK4* (Cox & Matthews, 2002) with a resolution of $2^{10}$, and datasets with lower resolutions are obtained by sub-sampling the highest resolution dataset.

For evaluation, firstly, we vary the total training samples ($N$) for all the operator models. We sample randomly and uniformly 5 times the training subset from the complete data $N = 1000$, and for each

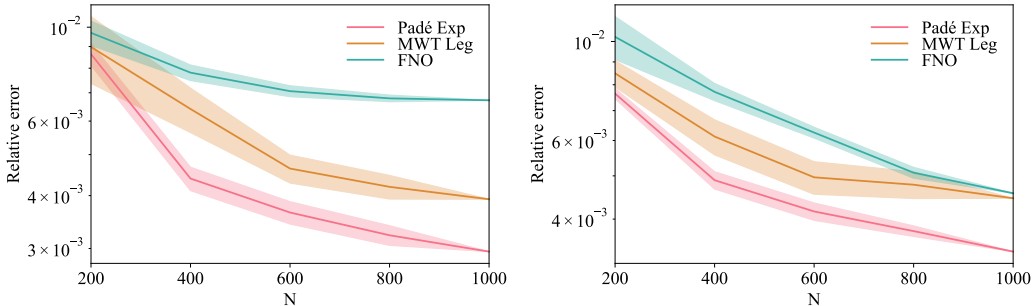

Figure 2: (Left) Number of training samples $N$ vs performance (relative L2 error) for neural operators evaluated on the KdV equation with s=1024. For $N < 1000$, each smaller dataset is sampled uniformly randomly 5 times from the complete dataset ($N = 1000$) and mean $\pm$ std.dev (shaded region) results are shown across the sampling experiments. (Right) Same analysis for KS equation with s=1024.

| Networks | s = 64 | s = 128 | s = 256 | s = 512 | s = 1024 |
|----------|--------|---------|---------|---------|----------|
| Padé Exp | **0.00301** | **0.00308** | **0.00311** | **0.00298** | **0.00295** |
| MWT Leg | 0.00372 | 0.00369 | 0.00391 | 0.00408 | 0.00392 |
| FNO | 0.00663 | 0.00676 | 0.00657 | 0.00649 | 0.00672 |
| MGNO | 0.12507 | 0.13610 | 0.13664 | 0.15042 | 0.13666 |
| LNO | 0.04234 | 0.04764 | 0.04303 | 0.04465 | 0.04549 |
| GNO | 0.13826 | 0.12768 | 0.13570 | 0.13616 | 0.12521 |

Table 2: Korteweg-de Vries (KdV) equation benchmarks for different input resolution $s$. The relative L2 errors are shown shown for each model.

sampling the models are evaluated. For consistency, each model is fed the same training sub-sample and the results are shown in Figure 2 (Left). We see that Padé Exp model has the sharpest decay compared to the other state-of-the-art neural operators on varying $N$. This shows that the proposed model is data-efficient and works well when less data is available. Next, we also evaluate the Padé Exp model on the complete data $N = 1000$ but by varying the input resolution as shown in Table 2. The proposed model performs consistently better for all the input resolutions.

## 3.2 KURAMOTO-SIVASHINSKY (KS) EQUATION

The Kuramoto-Sivashinsky (KS) equation is a fourth-order non-linear PDE derived to model the diffusive instabilities in a laminar flame front. For a given field $u(x, t)$, the KS PDE takes the following form:

$$
u_t = -u\frac{\partial u}{\partial x} - \frac{\partial^2 u}{\partial x^2} - \frac{\partial^4 u}{\partial x^4}, x \in (0, 1), t \in (0, 1]
$$

$$
u_0(x) = u(x, t = 0).
$$

(11)

The KS equation is also time-advection and according to the Table 1 notations, we have: $\mathcal{L} = -\frac{\partial^2}{\partial x^2} - \frac{\partial^4}{\partial x^4}$, $f(u) = -0.5u^2$, and $\mathcal{N} = \frac{\partial}{\partial x}$. The neural operator learns the mapping of the initial condition $u_0(x)$ to the solutions $u(x, t = 1)$. Similarly to the KdV equation in Section 3.1, the initial condition is sampled from a Gaussian random field $u_0 \sim \mathcal{N}(0, 5^4(-\Delta + 5^2 I)^{-2.5})$ with periodic boundary conditions. The equation is numerically solved using *chebfun* package (Driscoll et al., 2014) with a resolution of $2^{10}$, and datasets with lower resolutions are obtained by sub-sampling the highest resolution data set.

We evaluate the proposed model on the KS equations and compare with the existing works in a similar experimental setup as for the KdV equation in Section 3.1. The reduced training experiment results are shown in Figure 2 (Right), where we have uniformly and randomly sub-sampled the training samples from the complete dataset. For consistency, we have evaluated all models on the same sub-sampled training set. We again observe that the proposed Padé Exp model with its compact structure has the steepest decay of relative L2 error compared to the recent neural operator works.

| Networks | MAE | | | Relative L2 error | Net. vs FC |
|---|---|---|---|---|---|
| | C | R | D | | |
| Padé Exp | **1219 ± 130** | **1752 ± 666** | **211 ± 31** | **0.0155 ± 0.0034** | 82.14% (+652K) |
| MWT Leg | 3554 ± 1157 | 2928 ± 1338 | 284 ± 209 | 0.0245 ± 0.0043 | 62.0% (+18M) |
| FNO 3D | 4213 ± 391 | 3391 ± 1233 | 592 ± 157 | 0.0301 ± 0.0045 | 54.0% (+1.02M) |
| LNO 3D | 28502 ± 12698 | 6586 ± 3442 | 1465 ± 965 | 0.1056 ± 0.0394 | -105.0% (+238K) |
| Neural ODE | 4339 ± 1174 | 3443 ± 1408 | 443 ± 192 | 0.0310 ± 0.0069 | 53.8% (+172K) |
| Seq2Seq | 2798 ± 456 | 3317 ± 1690 | 346 ± 83 | 0.0273 ± 0.0058 | 63.7%(+1.8M) |
| Transformer | 7087 ± 972 | 6613 ± 2853 | 1722 ± 320 | 0.0501 ± 0.0094 | 13.4% (+15.2K) |
| FC | 10305 ± 2818 | 5885 ± 1609 | 1634 ± 686 | 0.0609 ± 0.0111 | (37.2K) |

Table 3: COVID-19 prediction benchmarks for different networks using 10-fold resampling with mean ± std. dev. across folds. The Mean Average Error (MAE) is presented for Confirmed (C), Recovered (R), and Deaths (D) counts averaged across 7 days of prediction for 50 US states. The relative L2 error is the test error for each model. The last column compares each network vs FC in terms of the total MAE improvement and total model parameters difference.

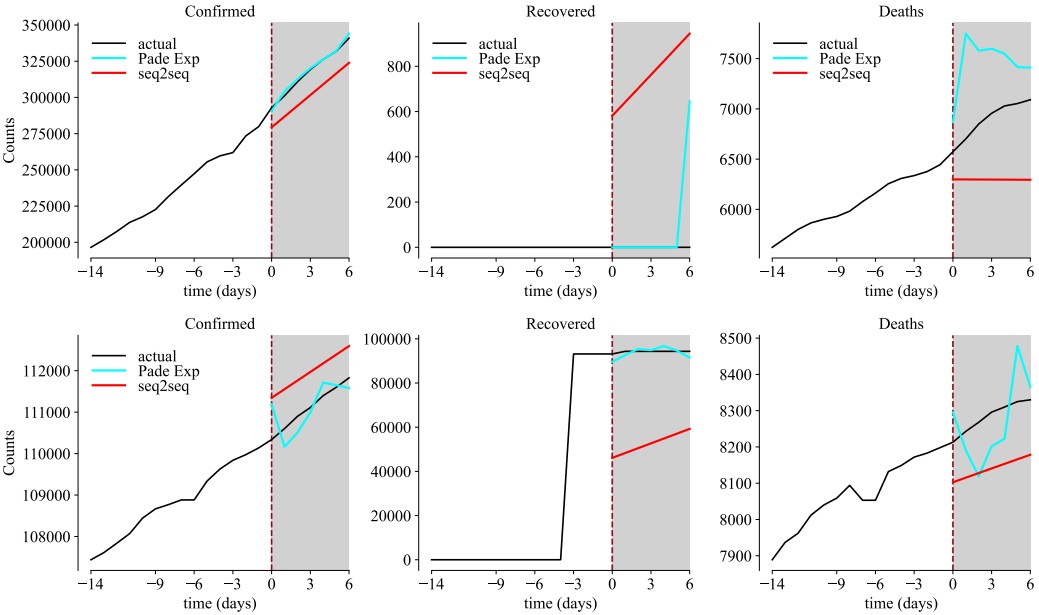

Figure 3: **COVID19 Forecasting**. Confirmed, Recovered, and Deaths count forecasting results for the 07/07/20 – 07/13/20 (chosen arbitrarily) using previous 2 weeks as the input. The Padé Exp prediction and the best non-neural operator scheme from Table 3 (seq2seq) is shown. **Top row**: Most populous US state California with population 39.77 M (2018 census). **Bottom row**: Same results for Massachusetts with a moderate population of 6.89 M.

By using the exponential operators, the IVPs can be efficiently solved as we witness from Figure 2. Finally, we also evaluate the proposed model for the complete data but varying the input resolution $s$ (see results in Table 10, Appendix B.4). We again conclude that the proposed model attains better performance than state-of-the-art approaches for all the resolutions.

## 3.3 EPIDEMIC FORECASTING: COVID-19 STUDY

The epidemic forecasting problem refers to the prediction of future counts of infected individuals, recovered individuals and deaths using the current observational data. A variety of compartmental models exist to model the epidemic spread with certain assumptions (see related work, Appendix A). The dynamics of the epidemic spread by modeling their time-evolution behavior is not exactly known, and may not apply to any problem, for example, the recent COVID-19 pandemic. Neural operators, providing a complete data-driven approach, are capable to learn *PDE agnostic maps*. Consequently, we show that the epidemic forecasting can be formulated as an operator map learning problem.

**Dataset** The COVID-19 data set[3] from April 12th 2020 to August 28th 2021 is provided by Johns Hopkins University (Dong et al., 2020). We take the data of 50 US States, and for each state, we have the total counts of daily reported confirmed (C), recovered (R), and deaths (D). We normalize the data of each state by their total population. Therefore, we have a daily collection of 2-dimensional data of size $50 \times 3$.

**Deep learning Benchmarks** In addition to the neural operators, we also compare against the state-of-the-art deep learning techniques: an auto-regressive fully connected (**FC**) network, Sequence to Sequence (**Seq2Seq**) (Salinas et al., 2019; Rangapuram et al., 2018) and **Transformers** (Vaswani et al., 2017) utilizing encoder-decoder structure. For Seq2Seq, we have used LSTM architecture for encoder and decoder. **Neural ODE** (Chen et al., 2019) utilizing latent ODE for time-series forecasting.

**Operator Map** The operator task is to learn the map between the 14 consecutive counts (C, R, D) to next 7 days data for each of the 50 US states. Let $d_t$ be the $50 \times 3$ array on day-$t$, then the operator map can be written as follows.

$$T(\underbrace{d_{-14}, d_{-13}, \ldots, d_{-1}}_{u_0(x)}) = (\underbrace{d_0, d_1, \ldots, d_6}_{u(\tau, x)}).$$

**Forecasting** The COVID19 forecasting benchmarks are presented in Table 3. Due to data scarcity (484 samples in total), we do a 10-fold resampling of the dataset to obtain train/test samples and the averaged results are presented for all models. We see that the proposed Padé exponential model achieves better performance than existing approaches especially in the presence of scarce and noisy data setup. The Seq2Seq performs best among all non-neural operator models. In terms of the total mean averaged error (MAE) (for C, R, and D counts), the Padé exponential achieves a $53\%$ improvement over the best neural operator (MWT Leg), and $52\%$ over the best non-neural operator model (Seq2Seq). The percentage improvement over the FC model and the difference between the total model parameters w.r.t. FC are shown in the last column. A sample forecasting is shown for 2 US states in Figure 3. We also show the corner cases (best and worst test sample) predictions in the Figure 8 (Appendix B.5) along with the best/worst prediction states.

## 4 ABLATION EXPERIMENTS

The ablation experiments are presented in the Supplementary Materials (Appendix B.1). The following experiments are performed; (*i*) Comparison between the proposed Padé approximation vs Taylor Series neural operator, (*ii*) the variations of $p, q$ for the $[p/q]$ Padé approximation model in Figure 1, and (*iii*) variation of the non-linearity module in the Padé neural model.

## 5 FUTURE DIRECTIONS AND CONCLUSION

Time-evolution analysis of PDEs and forecasting the future states from a set of observations is fundamental for studying a wide range of complex systems in physics, chemistry, geoscience, neuroscience, system biology, social, political, and climate sciences. In many such contexts, we need to solve an initial value problem consisting of a first-order time-evolution of several nonlinear spatial operators. To efficiently solve these initial value problems while also overcoming data science challenges (e.g., scarcity in the number of samples, samples corrupted by unknown noise types and sources), we proposed a combined multiwavelet and Padé approximation based exponential neural operator architecture. The proposed model has order-magnitude fewer parameters (see Table 3) and attains data-efficiency.

The IVPs deal with first-order time evolution, and thus, the exponential operator appears. In a more general setup, the higher-order time derivatives, or even fractional time derivatives should be considered to model the non-Markovian dynamics. In such cases, Reisz transform like approach helps. This is an interesting future direction for neural operator design, where data helps identifying the time-evolution order.

---

[3]`https://github.com/CSSEGISandData/COVID-19`

ACKNOWLEDGEMENT

We are thankful to the anonymous reviewers for providing their valuable feedback which improved the manuscript. We gratefully acknowledge the support by the National Science Foundation Career award under Grant No. CPS/CNS-1453860, the NSF award under Grant CCF-1837131, MCB-1936775, CNS-1932620, the U.S. Army Research Office (ARO) under Grant No. W911NF-17-1-0076, the Okawa Foundation award, the Defense Advanced Research Projects Agency (DARPA) Young Faculty Award and DARPA Director Award under Grant No. N66001-17-1-4044, an Intel faculty award, a Northrop Grumman grant, and Google cloud. A part of this work used the Extreme Science and Engineering Discovery Environment (XSEDE), which is supported by National Science Foundation grant number ACI-1548562. The third author has been supported in part by a NSF award under grant DMS-2108900 and by the Simons Foundation. The views, opinions, and/or findings contained in this article are those of the authors and should not be interpreted as representing the official views or policies, either expressed or implied by the Defense Advanced Research Projects Agency, the Army Research Office, the Department of Defense or the National Science Foundation.

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

| Networks | $\lambda = 0.05$ | $\lambda = 0.04$ | $\lambda = 0.03$ | $\lambda = 0.02$ |
|---|---|---|---|---|
| Padé Exp (p=5, q=6) | 0.00199 | 0.00237 | 0.00219 | 0.00215 |
| Padé Exp (p=4, q=2) | 0.00222 | 0.00231 | 0.00291 | 0.0159 |
| Taylor ($L_t$=6) | 0.00730 | 0.00690 | 0.00846 | 0.00704 |
| Taylor ($L_t$=4) | 0.00938 | 0.00842 | 0.00866 | 0.0193 |
| L'Hospital ($n$=6) | 0.00775 | 0.00699 | 0.00752 | 0.0194 |
| L'Hospital ($n$=4) | 0.00731 | 0.00798 | 0.00844 | 0.0199 |

Table 4: Taylor vs Padé Approximation for exponential operator evaluated on the KdV equation for s=1024 with varying input ($u_0(x)$) fluctuation strength $\lambda$ (low $\lambda$ implies high fluctuation). The highest polynomials degree are similar for each model by setting $L_t = \max(p, q)$ for Taylor and similarly $n = \max(p, q)$ for L'Hospital.

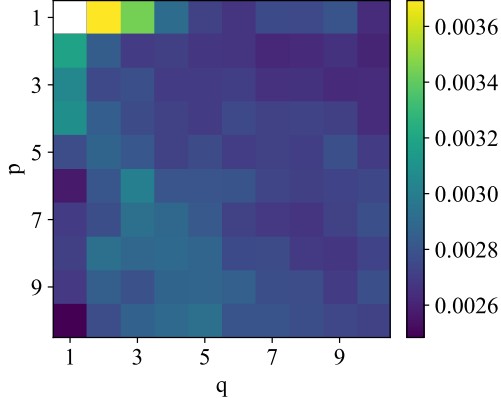

Figure 4: Polynomials degree $p, q$ grid variation for $[p/q]$ Padé neural model of Figure 1. The performance is evaluated on the KdV equation with s=1024.

## A    RELATED WORK

**Epidemic Forecasting**  In Epidemiology, the compartmental models are very common, where the population is assigned into discrete categories (for example, Susceptible, Infected, Recovered), also termed as SIR models which date back to 1927 (Kermack et al., 1991; Diekmann et al., 2021). Along the lines, several works exist that model the time-evolution of epidemic using various assumptions. The work of Chen et al. (2020) uses a ridge regression based prediction. An extension of SEIR (E for Exposed) compartments to sub-populations moving across different places in (Pei & Shaman, 2020). A convolution filter based approach in (Wang et al., 2020), where the total infected cases are modeled as convolution of novel cases with proportion of infected counts over time. A susceptible EIR model in (Zou et al., 2020b) incorporating the unreported cases. The work of Davis et al. (2020) which simultaneously models the transmission rate among US states. A physics-based model in (Wang et al., 2021a) that uses auto-differentiation using Runga-Kutta methods to estimate the model parameters.

## B    ADDITIONAL RESULTS

### B.1    ABLATION STUDY

**Taylor vs Padé Exponential Approximation**  We compare the Padé approximation based neural model in Figure 1 vs Taylor series approximation for evaluating the exponential of the operator $\mathcal{L}$. The Taylor series version is written as

$$e^{\mathcal{L}} \approx \sum_{j=0}^{L_t} \frac{1}{j!} \underbrace{(\mathcal{L} \circ \mathcal{L} \circ \ldots \circ \mathcal{L})}_{j-\text{times}},$$

where, $L_t$ is the truncated length of the series. We evaluate the truncated Taylor series approximation vs $[p/q]$ Padé approximation in Figure 1 for same number of maximum polynomial degree, i.e., $L_t = \max(p, q)$. For both models, we give input with varied degree of fluctuations (to make the data more challenging) and the output is governed by the KdV equation (see Section 3.1). The fluctuated

input is sampled from Random fields Filip et al. (2019) with controllable parameter $\lambda$ with lower values attaining higher fluctuations and the input $u_0(x)$ reaches the Brownian motion limit with $\lambda \to 0$. We see in Table 4 that the Padé approximation model works better compared to Taylor series based expansion for incorporating the exponential operators for each $\lambda$.

**L'Hospital Approximation** The exponential $\exp(x)$ can also be approximated as $\lim_{n \to \infty}(1+x/n)^n$ using L'Hospital rule (Krantz, 2004). For a given operator $\mathcal{L}$, the L'Hospital based exponential operator approximation can be written as

$$e^{\mathcal{L}} \approx \underbrace{(\mathcal{I} + \mathcal{L}/n) \circ (\mathcal{I} + \mathcal{L}/n) \circ \ldots \circ (\mathcal{I} + \mathcal{L}/n)}_{n-times},$$

where, $\mathcal{I}$ is the identity operator and $n$ is the length of truncation of operator compositions. We compare L'Hospital approximation with the Padé model in Table 4 for the same degree of operator polynomials, i.e., $n = \max(p, q)$. For the same experimental settings as taken in the Taylor approximation comparison, we see that the proposed Padé model performs better than the L'Hospital-based exponential approximation for all fluctuations strengths.

**Varying Padé polynomials degree** The numerator/denominator polynomials degree $p, q$ in eq. (5) for $[p/q]$ Padé approximation model in Figure 1 are hyper-parameters. We vary the $p, q$ over a 2D grid of $[1, 10] \times [1, 10]$ and evaluate for the KdV equation settings (s=1024) as mentioned in the Section 3.1 and the results are shown in Figure 4. We make the following observations, namely; (i) For $p = q = 1$, i.e., when there is no operator $\mathcal{L}$ and only the non-linear layer $\sigma(Wu + b)$, the model has the worst performance of 0.1063. This shows the importance of operator $\mathcal{L}$ in the Padé model. (ii) By increasing the $p, q$ in either direction, the performance improves as we attain better approximation with higher degree polynomials. The best performance is achieved as $(p, q) = (10, 1)$. (iii) The performance is roughly similar in the mid-range of $p$ and $q$'s, as long as $p > 1$ and $q > 3$ for the 1D experiment on KdV. Note that, although larger values of $p$ and $q$ are preferable but they also incur the additional cost of run-time which scales linearly with $p + q$ as evident from the recurrent structure in Figure 1.

**Structure of the non-linear MLP** We have used the non-linear layer $\sigma(Wu + b)$ in the Padé neural model for all the experiments with $\sigma(.)$=ReLU. Now, we vary the depth of this layer from 1 to 3 (each layer is followed by ReLU non-linearity) to see the individual contribution of this layer towards the final test performance. The experiments settings are same as for Table 2 and we fix $s = 1024$. Upon increasing the depth we do not see any improvements, in fact, after having three layers, the performance starts to degrade. A possible reason could be Denominator polynomial getting affected by the deeper non-linearity layer.

| Depth | L2 error |
|-------|----------|
| 1 | 0.00295 |
| 2 | 0.00339 |
| 3 | 0.00628 |

Table 5: Non-linear layer depth vs test error for the Padé neural model.

### B.2 PREDICTION AT HIGHER RESOLUTIONS

| Train \ Test | s = 2048 | s = 4096 | s = 8192 |
|-----------|----------|----------|----------|
| s=128 | 0.0423 (0.0473) | 0.0440 (0.0511) | 0.0450 (0.0544) |
| s=256 | 0.0229 (0.0315) | 0.0250 (0.0374) | 0.0263 (0.0427) |
| s=512 | 0.0124 (0.0230) | 0.0148 (0.0305) | 0.0162 (0.0372) |

Table 6: Padé exponential (MWT Leg) model trained at lower resolutions can predict the output at higher resolutions.

### B.3 TRAINING/EVALUATION WITH DIFFERENT $(p, q)$

Due to the recurrent structure of the Padé approximation model in Figure 1, with only trainable parameters $\{\theta_{\mathcal{L}}, W, b\}$, the model can be trained and tested on different values of $(p, q)$ by just varying the recurrence loop lengths of the numerator and denominator polynomial. We setup this experiment on the KdV equation (same setting as in Section 3.1). The Padé model is trained for $(p, q) = (5, 6)$ and $(4, 2)$ and then tested on $[1, 10] \times [1, 10]$ grid of $p, q$ tuples as shown in Figure 5.

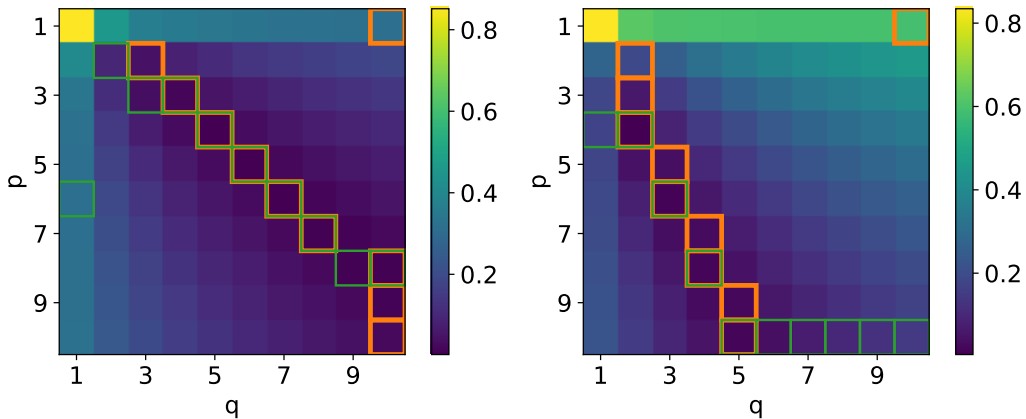

Figure 5: **Relative L2 error** when Padé model trained and tested on different $(p, q)$. (Left) The KdV data using $(p, q) = (5, 6)$ for training and other $(p, q)$ for testing. (Right) The KdV data using $(p, q) = (4, 2)$ for training and other $(p, q)$ for testing. For each plot, the Orange/Green color denotes best performance (lowest value) in the corresponding row/column, respectively.

We observe that the best performance (lowest relative L2 error) results when the train and test $(p, q)$ matches. In addition, we observe that for a fixed $q$, the best performance results when $p$ is such that p/q ratio is similar as the training ratio. For example, in Figure 5 (Left), we see that lowest value for each column is obtained when $p/q$ is closest to $5/6$ while in (Right) we see lowest value of each column when $p/q = 2$. The boundary cases diverge because either $p, q = 1$ does not have any operator $\mathcal{L}$ in the corresponding polynomial as evident from eq. (6). When trained for $(p, q) = (4, 2)$, the best performance for $q > 5$ saturates to $p = 10$ because of the maximum limit on the experimental grid ($p = 10$). Overall, this experiment suggests that the relative length of the polynomials during evaluation should be similar to training setting for best performance.

### B.4 KURAMOTO-SIVASHINSKY EQUATION

Comparison of relative error for different neural operators evaluated over KS equation is presented in Table 10.

### B.5 COVID-19 FORECASTING

Additional forecasting results for COVID-19 by considering the corner cases are presented in Figure 8.

### B.6 2D NAVIER-STOKES EQUATIONS

The Navier-Stokes (NS) equations are 2D time-varying PDEs describing the motion of viscous fluid substances. The NS equations can describe many physical processes and have wide range of practical uses. In this paper, to compare with the state-of-the-art models (Li et al., 2020a; Gupta et al., 2021) under the same conditions, we use the same data sets that have been published in (Li et al., 2020a), where the NS equations take the following form:

$$
\begin{aligned}
w_t(x, t) + u(x, t) \cdot \nabla w(x, t) - \nu \Delta w(x, t) &= f(x), & x \in (0, 1)^2, t \in (0, T] \\
\nabla \cdot u(x, t) &= 0, & x \in (0, 1)^2, t \in [0, T] \\
w_0(x) &= w(x, t = 0), & x \in (0, 1)^2
\end{aligned}
\tag{12}
$$

where $u$ is the velocity, $w$ is the vorticity such that $w = \nabla \times u$. The incompressible flow is modeled via divergence condition as $\nabla \cdot u(x, t) = 0$. We set the experiments to let the neural operator map the first 10 time units to last $T - 10$ time units of vorticity $w$. The initial condition is generated in Gaussian random fields according to $w_0 \sim \mathcal{N}(0, 7^{\frac{3}{2}}(-\Delta + 7^2 I)^{-2.5})$ with periodic boundary conditions and the forcing function is $f(x) = 0.1(\sin(2\pi(x_1 + x_2)) + \cos(2\pi(x_1 + x_2)))$. We experiment with different viscosities $\nu$, final time $T$, and the number of training pairs $N$: $(i)$ $\nu = 1e - 3, T = 50,$

| Networks | $\nu = 1e-3$ $T = 50$ $N = 1000$ | $\nu = 1e-4$ $T = 30$ $N = 1000$ | $\nu = 1e-4$ $T = 30$ $N = 10000$ | $\nu = 1e-5$ $T = 20$ $N = 1000$ |
|---|---|---|---|---|
| Padé Exp | 0.00621 | 0.1427 | 0.0619 | 0.1533 |
| MWT Leg | 0.00625 | 0.1518 | 0.0667 | 0.1541 |
| FNO-3D | 0.0086 | 0.1918 | 0.0820 | 0.1893 |
| FNO-2D | 0.0128 | 0.1559 | 0.0973 | 0.1556 |
| U-Net | 0.0245 | 0.2051 | 0.1190 | 0.1982 |
| TF-Net | 0.0225 | 0.2253 | 0.1168 | 0.2268 |
| Res-Net | 0.0701 | 0.2871 | 0.2311 | 0.2753 |

Table 7: Navier-Stokes Equation validation at various viscosities $\nu$ and prediction horizon $T$.

| Networks | N = 200 | N = 400 | N = 600 | N =800 | N = 1000 |
|---|---|---|---|---|---|
| Padé Exp | **0.00864±5.1e-4** | **0.00439± 2.8e-4** | **0.00365± 2.2e-4** | **0.00322± 1.7e-4** | **0.00295** |
| MWT Leg | 0.00898±16.1e-4 | 0.00641±7.7e-4 | 0.00463±3.5e-4 | 0.00420±2.7e-4 | 0.00392 |
| FNO | 0.00970±6.4e-4 | 0.00781±3.3e-4 | 0.00706±2.1e-4 | 0.00679±1.2e-4 | 0.00672 |

Table 8: Korteweg-de Vries (KdV) equation benchmarks for different numbers of training samples $N$. Top: Our method. Bottom: the state-of-the-art methods.

| Networks | N = 200 | N = 400 | N = 600 | N =800 | N = 1000 |
|---|---|---|---|---|---|
| Padé Exp | **0.00764±2.0e-4** | **0.00489± 2.2e-4** | **0.00416± 1.8e-4** | **0.00376± 1.0e-4** | **0.00338** |
| MWT Leg | 0.00849±5.7e-4 | 0.00612±5.5e-4 | 0.00496±4.2e-4 | 0.00478±3.3e-4 | 0.00445 |
| FNO | 0.01024±1.1e-3 | 0.00771±3.4e-4 | 0.00625±1.7e-4 | 0.00508±1.4e-4 | 0.00457 |

Table 9: Kuramoto–Sivashinsky (KS) equation benchmarks for different numbers of training samples $N$. Top: Our method. Bottom: the state-of-the-art methods.

$N = 1000$; $(ii)$ $\nu = 1e-4, T = 30, N = 1000$; $(iii)$ $\nu = 1e-4, T = 30, N = 10000$; $(iv)$ $\nu = 1e-5, T = 20, N = 1000$ on a $64 \times 64$ grid.

The time-varying 2D data is modeled as 3D operator learning problem. We implemented the Padé exponential model by taking $\mathcal{L}$ as 2-layered 3D CNNs with ReLU non-linearity and $p = q = 4$ in Figure 1. A total of 4 layers of multi-wavelet skeleton (Figure 7) with $k = 3$ are concatenated using ReLU non-linearity. The results are reported in Table 7, and we observe that Padé model performs well compared to the recent state-of-the-art neural operator approaches. For the less data setup of $N = 1000$ with $\nu = 1e-4$, we see in Figure 6 that the Padé Exp model quickly converges to the lowest value compared to the other neural operator approaches.

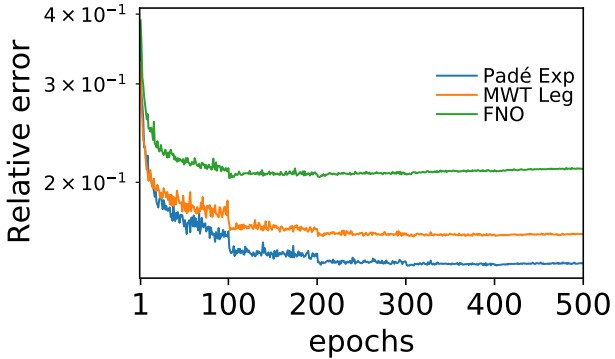

Figure 6: Relative $L2$ error vs epochs for MWT Leg with different number of OP basis $k$.

### B.7 Numerical Values

The numerical values for the training samples variation experiment for the KdV and KS equation in the Figure 2 are shown in Table 8 and Table 9, respectively.

| Networks | s = 64 | s = 128 | s = 256 | s = 512 | s = 1024 |
|---|---|---|---|---|---|
| Padé Exp | **0.00359** | **0.00326** | **0.00347** | **0.00348** | **0.00338** |
| MWT Leg | 0.00445 | 0.00414 | 0.00436 | 0.00485 | 0.00445 |
| FNO | 0.00461 | 0.00451 | 0.00469 | 0.00491 | 0.00457 |
| MGNO | 0.10362 | 0.12038 | 0.13361 | 0.13343 | 0.13799 |
| LNO | 0.04133 | 0.04020 | 0.04498 | 0.04341 | 0.04360 |
| GNO | 0.14037 | 0.14277 | 0.13862 | 0.14525 | 0.14363 |

Table 10: Kuramoto-Sivashinsky (KS) equation benchmarks for different input resolution $s$. The relative L2 errors are shown shown for each model.

## C NOTATIONS

### Operator Learning

| | |
|---|---|
| $T, \mathcal{L}$ | Operators between function spaces |
| $\mathcal{H}^{s,p}$ | Sobolev spaces such that constituent functions and their weak derivatives upto order $s$ have finite $L^p$ norms |

### Multiwavelets

| | |
|---|---|
| $\bigoplus$ | Subspace addition |
| $\mathbf{V}_n^k$ | $\{f \mid f$ are polynomials of degree $< k$ defined over interval $(2^{-n}l, 2^{-n}(l+1))$ for all $l = 0, 1, \ldots, 2^n - 1$, and assumes 0 elsewhere$\}$ |
| $\mathbf{W}_n^k$ | Orthogonal space to $\mathbf{V}_n^k$ such that $\mathbf{W}_n^k \bigoplus \mathbf{V}_n^k = \mathbf{V}_{n+1}^k$ |
| $P_n$ | Projection operator such that $P_n : \mathcal{H}^{s,2} \rightarrow \mathbf{V}_n^k$ |
| $Q_n$ | Projection operator such that $Q_n : \mathcal{H}^{s,2} \rightarrow \mathbf{W}_n^k$ |
| $L$ | Coarsest scale of the multiwavelet transform |

### Padé Approximation

| | |
|---|---|
| $\|\cdot\|$ | Operator norm |
| $\|\cdot\|_2$ | Euclidean L-2 norm |
| $[p/q]f$ | Padé approximation of function $f$ with numerator/denominator polynomial degree p/q, respectively |
| $\theta_\mathcal{L}$ | Parameters used to represent the operator $\mathcal{L}$ |
| $\theta_n$ | Number of parameters used to represent the operator $\mathcal{L}$ |
| $x \mapsto y = F(x; \Theta)$ | Mapping from $x$ to $y$ using function $F$ with parameters set $\Theta$ |

## D MULTIWAVELET EXPONENTIAL OPERATOR ARCHITECTURE

The Padé neural operator based multiwavelet transform model is shown in Figure 7. The input to the model is $s^{(n+1)}$, where $n = \log(M)$, and the output is $U_s^{(n)}$, where $n$ is the finest scale (or the log of input resolution). For a detailed description of multiwavelet transform, we refer the reader to (Gupta et al., 2021).

## E PROOF OF THEOREM 1

First note that, for the Padé approximation $[p/q]e^x$, the polynomial coefficients in eq. (5), we have

$$a_j = \frac{1}{j!} \frac{p(p-1)\cdots(p-j+1)}{(p+q)(p+q-1)\cdots(p+q-j+1)} \leqslant \frac{1}{j!} \left(\frac{p}{p+q}\right)^j. \tag{13}$$

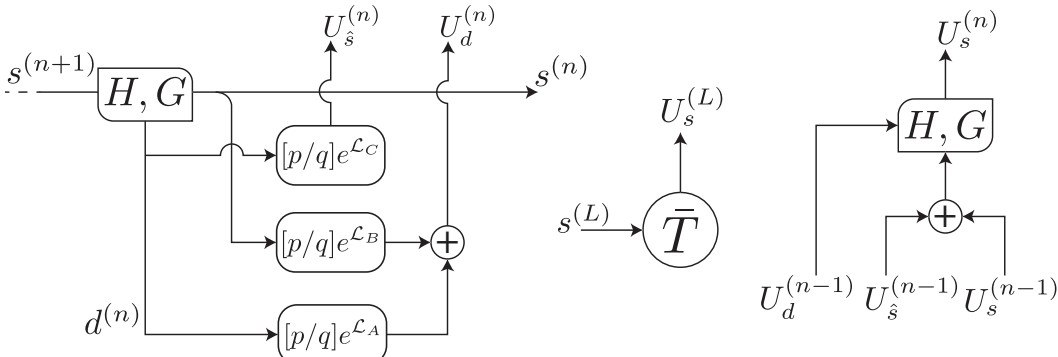

Figure 7: **Multiwavelet Padé Exponential Model**. The Padé exponential neural operator from Figure 1 is used in the skeleton of the multiwavelet transform based neural operator model. The inputs and outputs are recursively updated using the decomposition cell (Left) and reconstruction cell (Right).

and similarly,

$$|b_j| \leqslant \frac{1}{j!} \left( \frac{q}{p+q} \right)^j .$$

For the simplicity of notation: let $A(z) = A_{pq}(z) = \sum_{j=0}^{p} a_j z^j$, $A'(z) = \sum_{j=1}^{p} j a_j z^{j-1}$, $C(z) = \sum_{j=0}^{q} |b_j| z^j$, $C'(z) = \sum_{j=1}^{q} j |b|_j z^{j-1}$. Note $|B_{pq}(z)| \leqslant C(|z|)$ and $|B'_{pq}(z)| \leqslant C'(|z|)$. Using eq. (13), we can further write that

$$|A(z)| \leqslant \exp \left( \frac{p}{p+q} |z| \right), \quad |A'(z)| \leqslant \frac{p}{p+q} \exp \left( \frac{p}{p+q} |z| \right) \tag{14}$$

$$|C(z)| \leqslant \exp \left( \frac{q}{p+q} |z| \right), \quad |C'(z)| \leqslant \frac{q}{p+q} \exp \left( \frac{q}{p+q} |z| \right) \tag{15}$$

Further let $d$ and $n_\theta$ denote the dimensions of $x$ and $\theta$ respectively: $x, y, u, v \in \mathbb{R}^d$ and $\theta := \theta_{\mathcal{L}} \in \mathbb{R}^{n_\theta}$. For the operation $x \mapsto y$, we denote $y = F(x; \theta_{\mathcal{L}}, W, b) := [p/q]e^{\mathcal{L}}(x)$. Then $y = A(\mathcal{L})v$, $v = \sigma(Wu + b)$, $u = B(\mathcal{L})x$, and

$$\left\| \frac{\partial y}{\partial \theta_{\mathcal{L}}} \right\| = \max_{g \in \mathbb{R}^{n_\theta}, \|g\|_2 = 1} \left\| \lim_{t \to 0} \frac{1}{t} \left( F(x; \theta + tg, W, b) - F(x; \theta, W, b) \right) \right\|_2$$

$$= \max_{\substack{g \in \mathbb{R}^{n_\theta}, \|g\|_2 = 1 \\ h \in \mathbb{R}^d, \|h\|_2 = 1}} |\langle \nabla_\theta \langle y, h \rangle, g \rangle| \tag{16}$$

All matrix norms are spectral (i.e., the largest singular value), unless otherwise indicated as in equation (25), whereas all vectors norms are Euclidean (i.e., the L2-norm). Denote by $\partial_{g,\theta} = \sum_{j=1}^{n_\theta} g_j \frac{\partial}{\partial \theta_j}$, the differential operator induced by $g$. The quantity of interest is $|\partial_{g,\theta} \langle y, h \rangle|$. Next, by chain rule

$$\partial_{g,\theta} \langle y, h \rangle = \langle (\partial_{g,\theta} A(\mathcal{L})) v, h \rangle + \langle A(\mathcal{L}) \partial_{g,\theta} v, h \rangle$$

$$= \langle (\partial_{g,\theta} A(\mathcal{L})) v, h \rangle + \langle A(\mathcal{L}) DW \partial_{g,\theta} B(\mathcal{L}) x, h \rangle \tag{17}$$

where $D_g$ is a diagonal matrix of 1's and 0's depending upon the signatures of entries in $Wu + b$ (see for instance the computation of Lipschitz constants in Zou et al. (2020a)). Hence $\|D_g\| = 1$. Since $\sigma$ (ReLU) is contractive with Lipschitz constant 1, $\|v\|_2 \leqslant \|W\| \|u\|_2 + \|b\|_2 \leqslant \|W\| \|B(\mathcal{L})\| \|x\|_2 + \|b\|_2$. Substituting back in eq. (17) we obtain

$$|\partial_{g,\theta} \langle y, h \rangle| \leqslant \|\partial_{g,\theta} A(\mathcal{L})\| \|W\| \|B(\mathcal{L})\| \|x\|_2 + \|\partial_{g,\theta} A(\mathcal{L})\| \|b\|_2$$

$$+ \|A(\mathcal{L})\| \|W\| \|\partial_{g,\theta} B(\mathcal{L})\| \|x\|_2 \tag{18}$$

Using eq. (14)-(15), the spectral norms are bounded further by

$$\|A(\mathcal{L})\| \leqslant A(\|\mathcal{L}\|) \leqslant \exp \left( \frac{p}{p+q} \|\mathcal{L}\| \right), \tag{19}$$

$$\|B(\mathcal{L})\| \leqslant C(\|\mathcal{L}\|) \leqslant \exp \left( \frac{q}{p+q} \|\mathcal{L}\| \right). \tag{20}$$

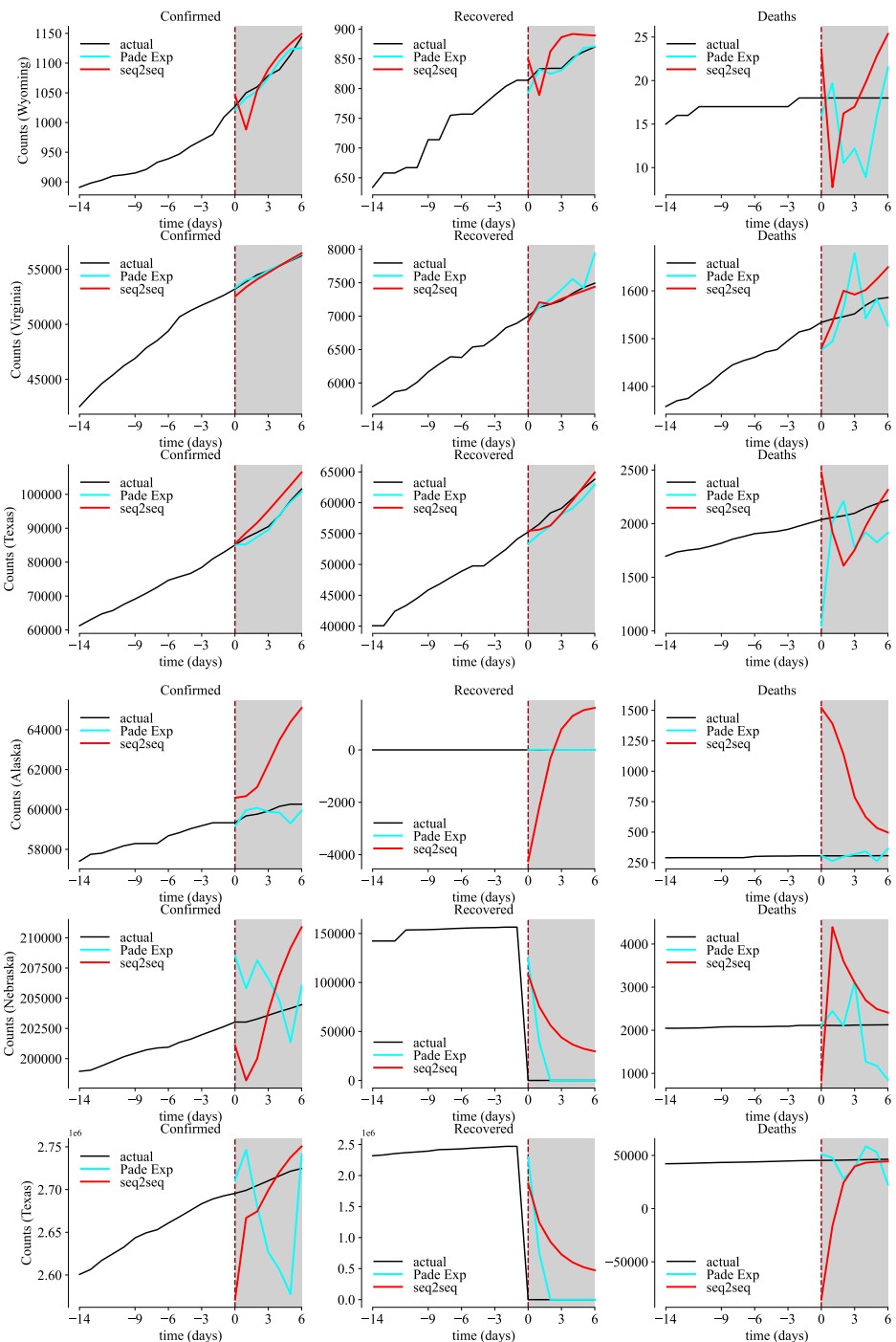

Figure 8: **COVID-19 Forecasting**. The Padé Exp prediction and the best non-neural operator scheme from Table 3 (seq2seq) is shown. **(Top 3 rows)** Prediction for the week 06/12/20 – 06/18/20 using previous 2 weeks data. This week is the test sample with least MAE averaged across Confirmed, Recovered, and Death counts for **Padé Exp**; or the test sample with best averaged prediction. Next, row 1-3 are US states in the order of best-middle-worst averaged prediction for this test sample. **(Bottom 3 rows)** Same analysis for the week 03/07/21 – 03/13/21 which is having largest MAE, or the test sample with worst averaged prediction. Similarly, row 4-6 are US states in the order of best-middle-worst averaged prediction for this worst test sample. The missing values are substituted with zeroes which is one of the reason for higher errors.

Next, by using the product rule, we obtain

$$\|\partial_{g,\theta}A(\mathcal{L})\| \leqslant \sum_{j=1}^{p} |a_j| \|\partial_{g,\theta}(\underbrace{\mathcal{L} \circ \mathcal{L} \circ \ldots \circ \mathcal{L}}_{j-\text{times}})\|$$

$$\overset{(a)}{\leqslant} \sum_{j=1}^{p} j a_j \|\mathcal{L}\|^{j-1} \|\partial_{g,\theta}\mathcal{L}\|$$

$$= A'(\|\mathcal{L}\|)\|\partial_{g,\theta}\mathcal{L}\|$$

$$\leqslant \frac{p}{p+q} \exp\left(\frac{p}{p+q}\|\mathcal{L}\|\right) \|\partial_{g,\theta}\mathcal{L}\|. \tag{21}$$

Similarly,

$$\|\partial_{g,\theta}B(\mathcal{L})\| \leqslant C'(\|\mathcal{L}\|)\|\partial_{g,\theta}\mathcal{L}\| \leqslant \frac{q}{p+q} \exp\left(\frac{q}{p+q}\|\mathcal{L}\|\right) \|\partial_{g,\theta}\mathcal{L}\|, \tag{22}$$

$$\|\partial_{g,\theta}\mathcal{L}\| \leqslant \sum_{j=1}^{n_\theta} |g_j| \|\frac{\partial\mathcal{L}}{\partial\theta_j}\| \leqslant \|g\|_2 \left(\sum_{j=1}^{n_\theta} \|\frac{\partial\mathcal{L}}{\partial\theta_j}\|^2\right)^{1/2}. \tag{23}$$

Using eq. (18), (22), and (23), we obtain the bound for eq. (16) as

$$\left\|\frac{\partial y}{\partial\theta_\mathcal{L}}\right\| \leqslant \left(\exp(\|\mathcal{L}\|)\|W\|\|x\|_2 + \frac{p}{p+q}\exp\left(\frac{p}{p+q}\|\mathcal{L}\|\right)\|b\|_2\right)\left(\sum_{j=1}^{n_\theta}\|\frac{\partial\mathcal{L}}{\partial\theta_j}\|^2\right)^{1/2}, \tag{24}$$

which is used to write the eq. (7) in Theorem 1. Similarly, the analysis is extended as

$$\left\|\frac{\partial y}{\partial W}\right\| = \max_{\Phi\in\mathbb{R}^{d\times d},\|\Phi\|_2=1}\left\|\lim_{t\to 0}\frac{1}{t}(F(x;\theta,W+t\Phi,b)-F(x;\theta,W,b))\right\|_2$$

$$= \max_{\Phi\in\mathbb{R}^{d\times d},\|\Phi\|_2=1}\|A(\mathcal{L})D_\Phi\Phi B(\mathcal{L})x\|_2 \leqslant \|A(\mathcal{L})\|\|B(\mathcal{L})\|\|x\|_2, \tag{25}$$

and

$$\left\|\frac{\partial y}{\partial b}\right\| = \max_{h\in\mathbb{R}^d,\|h\|_2=1}\left\|\lim_{t\to 0}\frac{1}{t}(F(x;\theta,W,b+th)-F(x;\theta,W,b))\right\|_2$$

$$= \max_{h\in\mathbb{R}^d,\|h\|_2=1}\|A(\mathcal{L})D_h h\|_2 \leqslant \|A(\mathcal{L})\|, \tag{26}$$

where, $D_\Phi$ and $D_h$ are diagonal matrices with 1's and 0's. Using eq. (25) and (26), we obtain the eq. (8) and (9) in the Theorem 1, respectively. $\square$

