# OpenReview forum: "Non-Linear Operator Approximations for Initial Value Problems"
_ICLR.cc/2022/Conference — ICLR 2022 Poster_

### Official Review · Reviewer_tvEi · 2021-10-31

**Correctness:** 4
**Technical Novelty And Significance:** 3
**Empirical Novelty And Significance:** 3
**Recommendation:** 8
**Confidence:** 4

**Main Review:**

This paper uses a rational polynomial to approximate the exponential function and proposed a Pade network. The idea is reasonable and the experimental result is convincing. Here are several minor points.
- The Pade approximation need the inverse, the paper use  a non-linear layer which we implement as a simple fully connected layer to approximate the inverse. Can the author justify this approximation, as an example justify the network is powerful enough?
-exp(x) can also be approximated by (1+(1/n)x)^{n}, can the author justify the benefit of the Pade approximation beyond the simple (1+(1/n)x)^{n} approximation. (the approximation doesn't need an inverse) I guess [1] is doing such approximation.
- After learning the operators in the Pade approximation, can the learned operator be generalized to other selections of (p,q)? As an example using one set of (p,q) for training and another (p,q) for testing?
- To justify an operator is learned, an experiment to show the generalization ability cross different grids is needed.

[1] Zongyi Li, Nikola Kovachki, Kamyar Azizzadenesheli, Burigede Liu, Kaushik Bhattacharya, Andrew
Stuart, and Anima Anandkumar. Markov neural operators for learning chaotic systems, 2021.

**Summary Of The Paper:**

This paper proposed a recurrent Pade network fro learning non-linear operator approximations for IVP.  The Pade exponential operator uses a recurrent structure with shared parameters to model the non-linearity compared to recent neural operators that rely on using multiple linear operator layers in succession. The paper showed that Pade network does not suffer from the issue of gradient explosion and the boundedness of the gradients can be established

**Summary Of The Review:**

My first main concern is that exp(x) can also be approximated by (1+(1/n)x)^{n}, the paper should take this as an baseline. Second is the nonlinear layer is not powerful enough to approximate the inverse.

---

> ### Author Response · Authors · 2021-11-17
> **Response to the reviews**
>
> We thank the reviewer for the evaluation, and especially for some nice suggestions for the additional experiments! The point-wise response is as follows:
>
> 1. The results in Theorem-1 and experimental evaluation consider a single non-linearity layer. The proof of Theorem-1 can be easily extended to more layers, and for experimental evaluation, we have already performed an Ablation over the number of layers as presented in Section B.1 (last para). We found out that on increasing the number of layers there is a degradation in the performance. A possible reason is an increase in depth leading to diminishing gradients for the denominator polynomial. After experimenting on a number of datasets with very different underlying Physics we found out that a single-layered non-linearity network is most useful.
>
> Another version that is not considered in this work for implementing the inverse polynomials is via a feedback loop such that the numerator polynomial $A(L)$ is in the forwarding part while the denominator polynomial $-B(L)$ is in the feedback loop. The overall effect would be (I+B(L))^{-1}A(L) by simple algebra. Future work can also exploit this simple trick.
>
> 2. We have followed the reviewer’s suggestion and performed experiments with approximating the $e^{x} \approx (1+x/n)^{n}$ while varying n. We have already compared our proposed Pade model with the Taylor series expansion version of the exponential in Ablation experiments Section B.1 Table-5. We have now updated Table-5 with this approximation (which we have referred to as L’Hospital), and we see that the Pade model is performing much better than already compared Taylor as well as this new L’Hospital based approximation for exponential. The suggested reference of Markov Neural operator is already cited as (Li et al. 2021) in the paper.
>
> 3. The reviewer has suggested an interesting experiment. The proposed model can indeed be trained and tested on different values of (p,q). We have performed an additional experiment in the revised version in Section B.3 and the results are presented in Figure-5. Specifically, we train our Pade model for the KdV equation (Section 3.1) for a given (p,q) (we tried (p,q)=(5,6) and (4,2)) and then tested on a 10x10 grid of (p,q) ranging from 1 to 10. We found out that the best performance results when train and test (p,q) are the same but interestingly we find that for achieving good results during evaluation we need to keep the relative length of the numerator/denominator polynomials similar to the training settings. This can be looked upon as the lower values of relative L2 error aggregating near the boundary where the ratio of p/q is similar to the training settings in Figure-5 for both cases.
>
> 4. Since the proposed Pade exponential model uses the same multiwavelet skeleton as in Ref (Gupta et al., 2021) of the paper, therefore, the architecture is independent of the input resolution. While we have already presented the results for different input resolutions in Table-2,3 but we have skipped the results of the train/test on different resolutions to avoid repetitions conveying the same arguments. Following the reviewer’s suggestion, we have added the following results as Table-7 in the revised version which shows the Pade model trained on lower resolutions can predict at higher resolutions as well.
>
> |             | s = 2048 | s = 4096 | s = 8192 |
> |---|---|---|---|
> |s = 128 | 0.0423 | 0.0440 | 0.0450 |
> |s = 256 | 0.0229 | 0.0250 | 0.0263 |
> |s = 512 | 0.0124 | 0.0148 | 0.0162 |

---

> > ### Comment · Reviewer_tvEi · 2021-11-20
> > **Thanks for the reivison**
> >
> > The revised version and added experiment is pretty interesting and impressive, I've raised my score to 8.

---

> > > ### Author Response · Authors · 2021-11-21
> > > **thank you!**
> > >
> > > We thank the reviewer for the updated score! and appreciate the nice suggestions that further enhanced the empirical results.

---

### Official Review · Reviewer_uucn · 2021-11-02

**Correctness:** 3
**Technical Novelty And Significance:** 3
**Empirical Novelty And Significance:** 2
**Recommendation:** 5
**Confidence:** 4

**Main Review:**

The idea in this paper is novel and promising. The approach follows the basics in PDEs, resulting in an efficient method.

There are multiple issues with this paper that I can not recommend acceptance at its current stage.
The first is related to the notation and explanation, and the second is the experimental study.

1) It seems there is room to improve notation and explanation.
1a) In eq 3, what does the O plus sign do?
1b)  Is L an operator between function spaces or finite spaces?
It is not clarified.
1c ) the problem construction is missing in the paper.

1d) how from Pade approximation on z in C do we have derivations of A applied on opeators?
1e) This paragraph "Pade ́Approximation Givenananalyticfunctionfpzq" needs clean up. The sentences are not easy to follow.

1f) what are the norms? what are the input-output spaces of operators? what are the spaces?
1g) In the proof of Theorem, A is applied on z and also L. Is A a spectral function?
1h) If everything here is finite-dimensional, then a more clarifying notation and problem structure are needed. It seems the authors jump between operator between function spaces and finite-dimensional vectors. The authors are encouraged to be consistent.
1i) what is F in the proof of the theorem?
1j) fix eq 19
1k) This list is longer, and the authors are encouraged to improve their problem construction.
1l) For what case Pade's approximation is valid? what are the a and b coefficients when dealing with operators?
2) The empirical study is not extensive compared to the prior works.
2a) are the numbers in the table and plots for one-step prediction? if yes, what happens if the operators are composed to predict future time steps?
2b) how the baselines are trained?
2c) Prior works study 2d Navier Stocks, the authors are encouraged to provide an extended study on that front to provide more compelling results on the benefits of their methods.
2d) study of further statistics is missing
3) Why not directly learn the exp(L) instead of using such thing as Pade approximation?


**Summary Of The Paper:**

This paper study the problem prediction in time-evolving partial differential equations.
Inspired by the nature of solutions in PDEs where the solution often time can be written in terms of exponent of the operator and Pade approximation exponent of operators, the authors propose a recurrent articture to learn solution operator in PDEs.

The paper proposes a nice idea and approach to solve the mentioned problem.

**Summary Of The Review:**

A great work, and a great idea.

However the notation needs a lot of work, the explanation needs a lot of work, the empirical study requires more work and explanation.

---

> ### Author Response · Authors · 2021-11-17
> **Response to the reviews**
>
> We are in debt to the reviewer for a careful read of our paper and are thankful for the kind words “great work”! We now take the opportunity to address the reviewer’s comments below.
>
> 1. For the future reader reference and following the reviewer’s suggestion we have added a detailed Table in the Appendix in the revised version outlining all of the notations used in this work.
>
> 1b. $\mathcal{L}$ is an operator between function spaces (Sobolev) as mentioned in Section-2.1 second para.
> 1d. The operator norm of $A(\mathcal{L})$ is upper bounded by $A(\Vert\mathcal{L}\Vert)$ because $A$ is a polynomial with non-negative coefficients. For a similar reason, the norm of $B(\mathcal{L})$ is controlled by $C(\Vert\mathcal{L}\Vert)$. These observations prove the left inequalities in (18) and (19). Inequalities in (20), (21), and (22) are obtained by further using the triangle inequality and the sub-multiplicative property of the operator norm, $\Vert TS\Vert\leq\Vert T\Vert \Vert S\Vert$, as appropriate.
> 1e. We have further clarified the text to better convey the mathematical notations.
> 1f. We assume that the reviewer is asking about Theorem-1 norms. As mentioned in the Theorem-1 statement, the norms in equation (7)-(9) are operator norms. In the Theorem-1 proof, the matrix norms are taken as Operator norms while vector norms are all Euclidean as mentioned in the paragraph following eq. (15).
> The input-output spaces for the operator problem are taken as Sobolev spaces as mentioned in Section-2.1 second para.
> 1g. In Theorem 1 of this work we assume $\mathcal{L}$ is a linear operator in which case $A$ plays the role of a spectral function. Thus, the spectrum of $A(\mathcal{L})$ is the same as the map $A$ applied on the spectrum of $\mathcal{L}$. We reference Section 2.3 in [R3.1], particularly to Theorem 2.3.2. More general, since Lipschitz maps $\mathcal{L}:V\rightarrow V$ so that $\mathcal{L}(0)=0$ forms a Banach algebra, Theorem 1 can be easily extended to (nonlinear) Lipschitz maps where the operator norm $\Vert\mathcal{L}\Vert$ is replaced by the Lipschitz constant Lip($\mathcal{L}$).
>
> 1h. We wish to clarify that the operators concerned in this work are between function spaces as mentioned in Section 2.1. The equation (4) provides a telescopic expansion of the finite rank approximation $P_{n}T P_{n}$ of operator $T$ -- note there was a typo in Eqn. 4 that we fixed.  In particular, $T$ may be an infinite rank operator. Theorem 1 applies to operators between function spaces.
> 1i. The function $F$ is used to denote the operation $x\mapsto y$. The notation is already mentioned in the Theorem-1 statement as $x\mapsto y = F(x;\theta_{\mathcal{L}},W,b)$. We will further add a similar statement in the Theorem-1 proof to improve its readability.
>
> 1j. We have fixed the typo.
>
> 1l. Pade approximation, as stated in Section-2.3 second para, is valid for any function that is analytic at 0. In this work, we are interested in the exponential functions, for which this condition is valid. As per the formulation, the coefficients are the same when we applied to operators.
>
> [R3.1] Barry Simon. A Comprehensive Course in Analysis, Part 4. Operator Theory, AMS 2015

---

> > ### Author Response · Authors · 2021-11-17
> > **other comments for experiments**
> >
> > 2. The previous works of Neural operators (references (Li et al., 2020a, Li et al. 2020b, Li et al., 2020a, Li et al. 2020c, Gupta et al. 2021)) have only used the simulated datasets such that the data is generated by some known physical phenomena. This is one of the criticisms faced by the previous works of being applied only to the academic datasets. However, in the current work, we have explored the true strength of the neural operator, i.e., by applying the neural operators to datasets with unknown underlying physical phenomena. In that context, we have explored the very recent COVID-19 data which has the obvious issues of (i) data scarcity, and (ii) noise due to improper daily reporting/collection. Through our extensive study in Section-3.2, we have shown that neural operators are very suitable for solving epidemic predictions without relying on any sub-optimal assumptions of compartmental models. Next, since we are concerned with the Initial Value Problems (IVP), we have also compared against the Korteweg-deVries (KdV) equation as well as studied a new equation Kuramoto-Shivasinsky (KS) which was not present in previous works. One motivation of the current work is to show the applicability of the exponential neural operators to real-world problems, and we note that the setting of epidemic prediction is exactly the same as 2D time-varying multi-step predictions. We have also presented the study on Navier-Stokes data in the revised version Section-B.5. There also we have observed that under the low-data setup, the Pade exponential model converges quickly to the lower values of L2 error compared to existing works.
> >
> > 2a. The operator learning problem maps the initial function $u_{0})(x)$ to a later function $u(x,\tau)$ as explained in Section-2.1. In Section-3.1 and 3.2, since we are mapping  $u_{0})(x)$ to $u(x,1)$, therefore, it is a mapping from t=0 to t=1. However, predictions for multiple time points can be simply obtained by expanding the dimensions of the input/output functions accordingly. In Section 3.3, we are doing predictions using multiple initial time points (14 days) and predicting multiple time points (7 days). Therefore, the input is expanded by 14 dimensions and output by 7 dimensions to model the function operator map problem (see Section 3.3 para 4 for details).
> >
> > 2b. The training of the baseline is explained in Section-3. All baseline neural operators, as well as the proposed model, are trained using the same training hyper-parameters.
> >
> > 2c. The Navier-Stokes dataset is also added in the Appendix Section-B.5.
> >
> > 2d. In Table-4, we have shown first and second-order statistics, since due to less data setup, we performed a 10-fold cross-validation re-sampling of the COVID19 data. Apart from only reporting the relative L2 error, like done in previous works of neural operators, we have also reported Mean Average Error (MAE) for the Epidemic study to better illustrate the outcomes of the different models. Similarly, for the case of less data setup for KdV and KS equations in Figure-2, we have shown first and second order statistics to show the variability due to sampling less data from the complete training set. The details are mentioned in Section-3.1 and 3.2. The numerical values of first and second order statistics are already reported as Table-9, 10 of the manuscript. For other experiments of Table-1,2, after training for 500 epochs, the performance is highly stable (due to sufficient training samples) which gets saturated much before 500 epochs as also observed in previous works of neural operators that used similar synthetic data. As an example, for 3 random seeds, the std. dev. for Pade Exp in Table-2 are 0.0000456 (s=64), 0.0000712 (s=128), 0.0000323 (s=256), 0.000009 (s=512), 0.0000112 (s=1024) which indicates high stability of values. The values can be reported in the revised version.

---

> > > ### Author Response · Authors · 2021-11-17
> > > **exponential clarification**
> > >
> > > 3. Learning directly the $\exp(\mathcal{L})$ has several issues, namely, (i) only in the event of $\mathcal{L}$ being a linear operator the $\exp(\mathcal{L})$ can be obtained computationally using the matrix exponential formulation. However, in the proposed formulation, the $\exp(\mathcal{L})$ using Pade approximation (Figure-1) does not put any such linearity restriction. In fact, for 2D datasets, we have used multi-layered Convolutional Neural Networks (CNN) with non-linearity. (ii) Even for the case of linear $\mathcal{L}$, for example, $\mathcal{L}$ being a convolutional operator (with fixed size $\theta_{\mathcal{L}}$), for a given input of size $n$ the required circulant matrix for the convolution has an impractical space complexity of $O(n^2)$ and hence exp(L) would also require unnecessarily $O(n^2)$ space. The computation of a circulant matrix exponential, even with the most efficient Pade approximation (Ref (Al-Mohy & Higham, 2009) of the paper) would require $O(n^3)$ computations. On the other hand, the proposed Pade model has $O(1)$ space complexity, i.e., size independent of the input size $n$- a necessary feature for resolution-independent architecture, and computations also take $O(n)$ for a fixed p,q.

---

### Official Review · Reviewer_QGHZ · 2021-11-02

**Correctness:** 3
**Technical Novelty And Significance:** 1
**Empirical Novelty And Significance:** 2
**Recommendation:** 3
**Confidence:** 4

**Main Review:**

The recently introduced neural operator framework exhibits several advantageous properties, and works that extend and generalize these methods are of interest. In this context, the current paper proposes to model the inherent nonlinearity in the data via exponential operators. From a differential equations viewpoint, this is a reasonable assumption as many IVPs could be solved using exponential maps. The theoretical stability guarantee is a nice addition, especially since RNN are known to be challenging to train. Finally, while the evaluation is not extensive with respect to state-of-the-art baselines and datasets, it does convey the message that the proposed method attains SOTA or potentially even beyond SOTA results.

My main concern regarding the paper and the reason for my lukewarm score is the unclear novelty in the method. Specifically, the expRNN (https://arxiv.org/abs/1901.08428) and dtriv (https://arxiv.org/abs/1909.09501) architectures considered the exponential map in the context of Lie groups. Essentially, these approaches use the scaling-squaring and a Pade approximant of Al-Mohy & Higham for which an analytic Jacobian can be derived and computed. In particular, no matrix inversion is explicitly calculated (see https://github.com/Lezcano/expRNN/blob/master/expm32.py). Also, this matrix exponential computation is *not* recurrent, and its stability is mostly governed by the modulus of the eigenvalues of the generating operator ($\mathcal{L}$ in your notation). Given the expRNN/dtriv line of work, it is not clear why the authors design a new recurrent architecture for computing the matrix exponential, and not simply using expm in torch. This approach will eliminate the recurrence, use less weights, will probably be faster, and will actually compute the matrix exponential. Regarding the last bit---since your approach is based on a learning component whose role is to learn the inverse polynomial, it is not entirely clear what is the actual space of operators your method learns. Please clarify.

Minor comments:
	- Is $\tau$ fixed in the operator problem? Why, and why is this problem so significant? Typically, one is interested in the prediction of one or more steps to the future (as in your COVID-19 example)
	- Is your $T$ finite- or infinite-dimensional? It seems like it is infinite-dimensional, and thus some discussion is required regarding its approximation as the $A_i, B_i, C_i$ are finite-dimensional.
	- I am somewhat puzzled about Eq. (4). Specifically, $T$ appears on the left and right sides of the equation. In addition, some of the operators are not specified ($P_i$?). How is this form used in practice?
	- Why $A_i, B_i, C_i$ are modeled as exponential operators?



**Summary Of The Paper:**

This paper introduces a new approach for solving the operator map problem which is based on the non-standard form and its approximation via exponential operators. The authors integrate their method in the recently introduced neural operator framework. The technique is evaluated on two synthetic PDE problems and one real life example. In all cases, the proposed approach improves over the other baselines on the relative L2 metric.

**Summary Of The Review:**

The proposed method is interesting as it extends a recently introduced framework (neural operators). However, the main technical novelty seems not necessary in the context of previous work.

---

> ### Author Response · Authors · 2021-11-17
> **Response to the reviews**
>
> We thank the reviewer for the feedback. The clarifications for the raised concerns are provided as follows.
>
> **Operator Exponential** The primary concern of the reviewer is to use the matrix exponential instead of the proposed method. We point-wise explain the issue with the suggested approach:
>
> 1a. For implementing $\mathcal{L}$ as a convolution operator as we did in the Experimental evaluation Section-3, for a given input of size $n$, the matrix formulation (circulant matrix for convolution operator) would require $O(n^2)$ space while the proposed model would only require $O(1)$ space for a fixed convolution kernel size $n_\theta \ll n$, and $p,q \ll n$ parameters. The matrix formulation would result in an impractical implementation for the neural operator. Note that a similar issue is also pointed-out in the reference (Hoogeboom et al. 2020) cited in  our paper where the exponential of convolution operator is used in the Generalized Sylvester flows. There the authors have used Taylor approximation, and we have already compared the proposed model with Taylor approximation in Ablation study Section-B.1 where Pade is ~5 times better than Taylor-based exponential.
>
> On a similar note, the computational complexity using matrix exponential would be $O(n^2)$ while the proposed Pade model induces $O(n)$ complexity for a fixed convolutional kernel size $n_\theta \ll n$ and fixed $p, q \ll n$.
>
> 1b. If the dynamics are non-linear, which are the cases that we have taken in Section-3, then the exponential matrix formulation would not be able to approximate the non-linear perturbations that are acting in the system. On the other hand, the Pade formulation for operator $\mathcal{L}$ in eq. (6) and then its implementation in Figure-1 does not put such restrictions of linear operator input. In fact, it is because of the non-linear operator $\mathcal{L}$, that we have implemented as multi-layered CNNs in Section-3, that we are able to learn the time-evolution operator using an exponential of non-linear maps. The canonical blocks offered by Figure-1 are expressive to learn the non-linearity in the underlying equations.
>
> 1c. Implementing the matrix exponential-based formulation would also tie the model to a certain resolution. The proposed Pade architecture using CNNs based architecture for $\mathcal{L}$ is a resolution-independent scheme, and therefore can be used at multiple resolutions, in other words, train/test at different resolutions.
>
> 2. Since the training is done by using the input and output of the neural architecture in Figure-1, we aim to learn a non-linearity layer that mimics the inversion of the polynomial $B(\mathcal{L})$. It cannot be guaranteed that the non-linear layer exactly does the inversion because learning minimizes the overall loss function of this architecture. Overall, the proposed architecture learns the time-evolution operator between the input and output.
>
> **Other comments** The use of $\tau$ is notational in the sense that we aim to learn the map from some initial condition $u_{0}(x)$ to values at a later time $u(x,\tau)$. We do not use explicitly $\tau$ anywhere in learning the operator map. This problem setup is mentioned as the Initial Value Problem (IVP) in Section 2.1. The multi-step prediction is an important problem (studied in Section 3.3 as epidemic prediction) as the reviewer has acknowledged. In such cases, the operator learning problem can be formulated by expanding the dimensions of the input and output accordingly. For example, in the COVID19 study, to predict the 7 days using the previous 14 days data, we expanded the dimensions of the input $u_{0}(x)$ by 14 and output $u(x,\tau)$ by 7, as mentioned in Section 3.3 para-4.
>
> The operator $T$ is indeed infinite-dimensional as formulated in Section 2.1. We have corrected the typo in eq. (4), where the right-hand side is the projection of $T$ onto ${\bf V}_{n}^{k}$ which we guess is the primary source of confusion here. The notations of $P_i, Q_i$ are also added. The eq. (4) is just a telescopic sum series to expand the projected version of $T$ onto multiple scale-decoupled components. The modeling of $A_i, B_i, C_i$ as Pade model is a design choice to better learn the time-evolution operator which is very useful in solving IVPs.

---

### Official Review · Reviewer_EkGn · 2021-11-03

**Correctness:** 4
**Technical Novelty And Significance:** 3
**Empirical Novelty And Significance:** 3
**Recommendation:** 6
**Confidence:** 4

**Main Review:**

The paper proposes a novel and natural model for operator learning.

Pros:
1. The paper writes out the time update as in Table 1.
2. The paper uses the Pade approximation for the exponential operator.
3. The paper gets the state-of-art results on both PDEs and real-life benchmarks.

Cons:
It seems to me that the exponential operator only addresses the linear term exp(tA), while the most challenging and interesting part of the non-linear equation is the time integral term addressing the non-linear term N. This design may have a limitation on harder non-linear equations such as the Navier-Stokes equation. Would it be possible to add an additional term to approximate the time integral?

**Summary Of The Paper:**

This paper proposes to learn the exponential operator in time evolution using the Pade approximation. The proposed method shows the states of art results on 1d Burgers, KdV equation, as well as the covid 19 benchmark.

**Summary Of The Review:**

In my opinion, this work is novel and meaningful. However, it is less satisfactory not to address the more interesting non-linear term N. I recommend borderline acceptance.

---

> ### Author Response · Authors · 2021-11-17
> **Response to the review**
>
> We thank the reviewer for acknowledging the novelty of the proposed work! The concern raised by the reviewer is addressed below.
>
> **Exponential operator** We wish to clarify that we have not imposed any restriction of linearity on the operator $\mathcal{L}$ while defining $\exp(\mathcal{L})$ in eq. (6) of the paper as well as its implementation in Figure-1. In fact, this is one of the strengths of using the proposed Pade architecture which can be easily used with non-linear operators, unlike matrix exponential which is by definition limited to the exponential of linear finite-dimensional operators. We will further add a brief clarifying sentence before eq. (6) to mention that $\mathcal{L}$ is non-linear in-general.
>
> In the context of the theoretical results, it is only for the sake of simplicity of the proof notations that we have taken the case of linear $\mathcal{L}$ in Theorem-1. However, the results of Theorem-1 can also be easily extended to the exponential of non-linear operator N by defining the norms as Lipschitz coefficients Lip(N) instead of Operator norms as taken in eq. (7)-(9) such that
> $$
> Lip(N) = \sup\limits_{x\neq y}\frac{\Vert N(x)-N(y)\Vert}{\Vert x-y\Vert}.
> $$
>
> Now, in Table-1 and footnote 1, the $\mathcal{L}$ is mentioned as a linear operator because of the way the time-advection equations are defined to isolate linearity with the perturbations of non-linearity. We wish to point out that for solving such equations, using time-discretization techniques like Exact Linear Part (ELP), after discretizing the integrals, a canonical function that emerges is: $Q_k(x) = (e^x - E_k(x))/k!$, where $E_k(x)$ is the truncated exponential up to $k$ Taylor terms (see reference (Beylkin et al. 1998) cited in  the paper). The final solution is a weighted linear combination of the function of operator $\mathcal{L}$ as $Q_k(\mathcal{L})$. Therefore, the time-evolution using operators can be learned by the exponential of non-linear operators. We have shown that such a canonical exponential of non-linear operators (Figure-1) is expressive enough to learn the non-linear perturbations present in the underlying PDEs.
>
> In the experimental evaluation (Section 3) using equations like Korteweg-deVries, Kuramoto-Shivasinsky as well as the real-world study of COVID19, we have taken multi-layered convolutional neural networks (CNNs) with non-linearities to model $\mathcal{L}$ as mentioned in Section 3 (model implementation). Although we intended to show the potential of neural operators over a challenging real-world study that has issues like data scarcity, noise such as the COVID19 case study we presented; we adopted a similar forecasting framework as for 2D time-varying dynamics where we obtained significant improvements over state-of-the-art neural operators as well as other deep learning benchmarks. Under a similar setup, we have also presented the results for 2D Navier-Stokes in the revised version in Section-B.5. Here, we see in Figure-6 that the Pade model quickly converges to its best performance compared to other models in the regime of fewer data.
>
> All in all, we have shown that for solving IVPs, the exponential of operators (possibly non-linear) is very useful and can be used as a basic block for learning time-evolution operators for complex setups (non-linear PDEs, multi-compartmental epidemic forecasting) with non-linear perturbations.

---

> > ### Comment · Reviewer_EkGn · 2021-11-25
> > **Ask for clarifications**
> >
> > I may miss some parts, so please correct me if I am wrong. Based on my understanding, given a dynamical system du/dt = F(u), the evolution operator G: u(t0) -> u(t) in general cannot be written as an exponential. Only if the system is a semigroup and the operator F is linear, G can take the form of an exponential. Especially, in Table 1, the equation holds only if L is the linear term and N is the non-linear term. We can denote L as the non-linear and compute exp(L), but that will be a linear approximation of the non-linear system. Does the author mean to approximate the general evolution operator instead of an exponential operator?

---

> > > ### Author Response · Authors · 2021-11-27
> > > **Author clarifications**
> > >
> > > The reviewer is correct in stating that the time-evolution operator for general IVPs cannot be written straightforwardly as exponential unless some assumptions like semigroup are taken. For more details, we refer to References (Pazy, 1983; Yoshida, 1989) of the paper. We wish to clarify that we have not relied on any such assumption in this work, as the purpose of the current work is to develop a neural architecture for the time-evolution operator.
> > >
> > > Note that, since the data available for learning is input $u_{0}(x)$ and output $u(\tau,x)$, we are indeed learning the time-evolution operator and not exactly an exponential operator. Now for that purpose, motivated by the exponential operators, we have proposed a Pade approximation-based neural architecture in Figure-1 for learning the time-evolution operator. We have shown that such neural architecture is indeed (i) capable of learning the non-linear time-evolution operator for various academic and real-world settings (Section-3), (ii) data-efficient because of its compact nature in terms of trainable parameters (Section-3.3), and (iii) has bounded gradients (Theorem-1).

---

### Decision · Program_Chairs · 2022-01-20

**Decision:**

Accept (Poster)

**Comment:**

The paper provides a unique contribution that uses Padde approximations to approximate non-linear operators for solving initial value problems in PDEs. The paper contains also a non-trivial experiment with a real-world dataset that showcase the impact of the proposed model. The authors have provided a strong rebuttal and therefore I recommend Accept.